# Quality Markers’ Discovery and Quality Evaluation of Jigucao Capsule Using UPLC-MS/MS Method

**DOI:** 10.3390/molecules28062494

**Published:** 2023-03-08

**Authors:** Yanmei He, Fangfang Wu, Zhien Tan, Mengli Zhang, Taiping Li, Aihua Zhang, Jianhua Miao, Min Ou, Lihuo Long, Hui Sun, Xijun Wang

**Affiliations:** 1National Chinmedomics Research Center, National TCM Key Laboratory of Serum Pharmacochemistry, Chinmedomics Research Center of State Administration of TCM, Laboratory of Metabolomics, Department of Pharmaceutical Analysis, Heilongjiang University of Chinese Medicine, Harbin 150036, China; 2National Engineering Laboratory for the Development of Southwestern Endangered Medicinal Materials, Guangxi Botanical Garden of Medicinal Plants, Nanning 500023, China

**Keywords:** serum pharmacochemistry, jigucao capsule, quality markers, ultra-high-performance liquid chromatography-mass spectrometry, quality evaluation

## Abstract

Jigucao capsules (JGCC) have the effects of soothing the liver and gallbladder and clearing heat and detoxification. It is a good medicine for treating acute and chronic hepatitis cholecystitis with damp heat of the liver and gallbladder. However, the existing quality standard of JGCC does not have content determination items, which is not conducive to quality control. In this study, serum pharmacochemistry technology and UNIFI data processing software were used to identify the blood prototype components and metabolites under the condition of the obvious drug effects of JGCC, and the referenced literature reports and the results from in vitro analysis of JGCC in the early stage revealed a total of 43 prototype blood components and 33 metabolites in JGCC. Quality markers (Q-markers) were discovered, such as abrine, trigonelline, hypaphorine and isoschaftoside. In addition, ultra-high-performance liquid chromatography–triple quadrupole mass spectrometry (UPLC-QQQ-MS) was used to determine the active ingredients in JGCC. The components of quantitative analysis have good correlation in the linear range with R^2^ ≥ 0.9993. The recovery rate is 93.15%~108.92% and the relative standard deviation (RSD) is less than 9.48%. The established UPLC-MS/MS quantitative analysis method has high sensitivity and accuracy, and can be used for the quality evaluation of JGCC.

## 1. Introduction

JGCC is derived from the traditional application of the folk herb *Abrus cantoniensis* Hance. The monarch medicine *Abrus cantoniensis* Hance in the prescription was first published in “Lingnan Herbs Collection” written by Budan Xiao [1], also called “Huangtou Herb” or “Dahuang Herb”. It has a long history of being used to treat jaundice [2,3]. JGCC is composed of 10 medicinal materials, including *Abrus cantoniensis* Hance, *Artemisia capillaris* Thunb, *Gardenia jasminoides* Ellis, *Panax notoginseng* (Burk.) F.H.Chen, *Paeonia lactiflora* Pall, *Origanum vulgare* L., *Ziziphus jujuba* Mill, *Lycium barbarum* L., *Sus scrofadomestica* Brisson and Bovis calculus Artifactus. Damp-heat jaundice syndrome (DHJS) is a common disease, also known as Yang Huang syndrome [4]. It is detailed in “Treatise on Typhoid and Miscellaneous Diseases”, written by Zhang Zhongjing. It mainly presents with three yellow symptoms, including yellow eyes, yellow body and yellow urine, as well as pathological manifestations such as thirst and fever, inappetence, disgusting vomits and poor defecation [5]. Metabolomics studies have shown that JGCC has a significant therapeutic effect on rats with DHJS by acting on some abnormal metabolic pathways such as pentose and glucuronate interconversions, arachidonic acid metabolism and primary bile acid biosynthesis [6,7]. On the other hand, modern pharmacological studies have shown that JGCC combined with entecavir antiviral treatment of patients with chronic hepatitis B can enhance its antiviral effects and delay the progression of liver fibrosis by reducing serum transforming growth factor-β1, transaminase, total serum bilirubin and other liver function indicators [8,9]. JGCC can also assist polyene phosphatidylcholine capsules in the treatment of patients with nonalcoholic fatty liver, and it can significantly reduce the degree of fatty liver and serum liver fibrosis indicators [10]. JGCC is a drug for the treatment of chronic cholecystitis [11] and acute hepatitis [12]. It can also reduce the serum levels of alanine aminotransferase (ALT), aspartate aminotransferase (AST) and malondialdehyde (MDA) and increase the levels of albumin, total protein and superoxide dismutase in immune hepatic fibrosis rats [12]. It can reduce the degree of pathological damage of liver tissue, and has obvious protective effects on immune liver fibrosis [13,14]. Therefore, JGCC is a good medicine for treating liver and gallbladder disease.

The serum pharmacochemistry technology of traditional Chinese medicine (TCM) is used to screen the pharmacodynamic material basis and determine the Q-markers of TCM/formula from the blood migrating components after oral administration, and it is known as one of the effective methods to solve the quality evaluation of TCM. Serum pharmacochemistry is based on classical pharmacochemical research methods and uses modern separation and multi-dimensional combined technologies to analyze, identify and characterize the transition components in the human/animal serum after oral administration of Chinese medicines. It is aimed to clarify the correlation between the activities and the traditional pharmacodynamic components of TCM in order to determine their pharmacodynamics material basis. It is also an applied science to study the processes of TCM in vivo [15] and is now recognized as a widely used method to study the pharmacodynamic material basis of Chinese medicine. The pharmacodynamic material basis of TCM refers to the chemical components contained in the TCM which can express the clinical efficacy of the drug; it is a key factor related to the quality issues such as the effectiveness and safety of TCM, and the technology and method for research and confirmation of pharmacodynamic material basis are the key scientific issues which restrict the modernization and international development of TCM [16]. Q-markers were first proposed in 2016 by Changxiao Liu, and they attracted great attention from academia and industry as soon as they were proposed [17]. Chinese medicine Q-markers are chemical substances that are inherent in Chinese herbal medicines and their products (including medicinal slices, Chinese medicine decoctions, Chinese medicine extracts and Chinese patent medicine preparations); they can be formed during processing and preparation and are closely related to the functional properties of Chinese medicines. Q-markers reflect the safety and effectiveness of TCM and are respected for quality control of the labeled substances. After several years of development and application [18,19,20,21,22], scientists and technicians have conducted a lot of research and exploration on Q-markers of TCM to make great progress in the quality research of Chinese medicines. A qualitative leap in the theory, idea and method of quality control of TCM is realized.

Although JGCC is effective in treating acute and chronic hepatitis and cholecystitis [8,9,11,12], there are only a few relevant reports on the material basis of its efficacy at home and abroad, and the quality control standard lacks content determination items. The current quality standard [23] includes a character description, using *Artemisia capillaris* Thunb as a reference material for thin-layer chromatography identification, stating that the moisture content should not exceed 7.0%. Supplementary research on the quality standards of JGCC is urgent. The purpose of this study is to identify the prototype components and metabolites of JGCC entering the blood under the condition of obvious drug effect by using the serum pharmacochemistry technology of TCM, so as to determine the Q-markers of JGCC for the treatment of DHJS and provide the basis for the selection of the content determination index of JGCC. In addition, UPLC-MS/MS is used for quantitative analysis of various effective components, so as to achieve the purpose of quality control and provide a theoretical basis for improving the quality standard of JGCC.

## 2. Results

### 2.1. The Evaluation for Preparation of DHJS Rat Model and Effectiveness of JGCC

The results of liver hematoxylin-eosin (HE) staining are shown in Figure 1. In the control group, hepatocytes were arranged radially and were orderly, the size of the central vein was normal and the morphology of hepatocytes was normal (Figure 1A). In the model group, inflammatory cells infiltrated the portal area, collagen fibers proliferated around the vessels, the vessel wall thickened and small bile duct-like epithelial cells proliferated and the number of small bile ducts increased (Figure 1B). The inflammatory cells in the JGCC group decreased significantly, and the structure of the hepatic lobule was similar to that in the control group, which was clear and complete (Figure 1C).

The results of bile duct HE staining are shown in Figure 1. In the control group, the cell structure of bile duct tissue was basically complete, and there was a single layer of bile duct epithelial cells without inflammatory cells (Figure 1D). In the model group, the bile duct epithelial cells were necrotic and exfoliated, small bile duct-like epithelial cells proliferated and the number of bile ducts were significantly increased, and inflammatory cells were infiltrating in the outer membrane of the bile duct (Figure 1E). In the JGCC group, only a small amount of bile duct epithelial cells degenerated, and the structure of bile duct tissue was basically complete (Figure 1F).

The biochemical indexes of rat serum were analyzed by an automatic biochemical analyzer, and the results are shown in Figure 2. Compared with the control group, AST, ALT, alkaline phosphatase (ALP), total bilirubin (T-Bili), γ-glutamyl transferase (γ-GT) and total bile acid (TBA) in the model group increased significantly. Compared with the model group, AST and γ-GT in the JGCC group showed a callback trend, while ALT, ALP, T-Bili and TBA significantly decreased, which tended towards being close to the control group.

In conclusion, the liver and bile duct of rats had been damaged, indicating that the rat model of DHJS had been successfully prepared, and JGCC had a good therapeutic effect on DHJS rats.

### 2.2. Prototype Components Analysis of JGCC Found in the Blood

In this study, on the basis of successfully establishing the rat model of DHJS, JGCC was orally administrated for treatment, and the blood components of JGCC were analyzed in a markedly effective state. UPLC-Q-TOF-MS was used to collect the profile data in serum samples of JGCC in the positive and negative ion modes, and the two preparation methods of serum samples were compared. The results showed that the numbers and shapes of the base peak intensity (BPI) chromatograms obtained from method 1 were better than those in method 2 (Figure 3). Therefore, method 1 was subsequently chosen as the preparation method for the serum samples. Using the UNIFI software, the reference compound fragments were compared and the prototype blood components of JGCC were characterized and identified.

Abrine was the main active ingredient in the monarch medicine *Abrus cantoniensis Hance*. The following is an example of abrine to explain the analysis process of the prototype blood composition of JGCC. In positive ion mode, after automatic noise reduction, the peak matching and peak extraction with the UNIFI software, an *m*/*z* of 219.11 was extracted at t_R_ = 3.21 min from the serum sample of the DHJS rats administered with JGCC (Figure 4A), which was consistent with the in vitro solution of JGCC. At the high and low energy channels (the high collision energy was 20–40 V and the low collision energy was 6 V), the parent ion peak was *m*/*z* 219.11 and the fragment ions *m*/*z* 188.07, 146.06 and 132.14 were scanned (Figure 4C,D). It was found that *m*/*z* 188.07 was obtained by dropping CH_5_N from *m*/*z* 219.11, *m*/*z* 146.06 was the result of dropping C_2_H_3_NO_2_ from *m*/*z* 219.11 and *m*/*z* 132.14 was obtained by *m*/*z* 219.11 dropping C_3_H_5_NO_2_ (Figure 5C)_._ The information of the parent and fragment ions mentioned above were found in the abrine standard (Figure 5A) and the serum samples of DHJS rats administered with JGCC (Figure 5B). This was consistent with the cleavage rules of abrine that were reported in the literature [24,25]. However, the above information was not found in the control and model groups (Figure 4B), so it was determined that abrine was a prototype blood component.

A total of 43 prototype blood components of JGCC were characterized and identified (Table 1), which were respectively derived from nine components of *Abrus cantoniensis* Hance, two components of *Artemisia capillaris* Thunb, six components of *Gardenia jasminoides* Ellis, five components of *Paeonia lactiflora* Pall, four components of *Panax notoginseng* (Burk.) F.H.Chen, four components of *Origanum vulgare* L., three components of *Ziziphus jujuba* Mill, four components of *Lycium barbarum* L., eight components of *Sus scrofadomestica* Brisson and six components of Bovis calculus Artifactus. Among them, there was a common component of *Abrus cantoniensis* Hance and *Origanum vulgare* L., and a common component of *Artemisia capillaris* Thunb and *Gardenia jasminoides* Ellis. *Sus scrofadomestica* Brisson and Bovis calculus Artifactus have four common components, and *Paeonia lactiflora* Pall, *Gardenia jasminoides* Ellis and *Origanum vulgare* L. have a common component. According to a large number of studies in the literature, 27 of the 43 prototype blood components, including trigonelline, 3,4,5-trihydroxybenzoic acid, geniposidic acid, abrine, chlorogenic acid, hypaphorine, *p*-coumaric acid, geniposide, genipin-1-gentiobioside, vicenin-2, albiflorin, isoschaftoside, paeoniflorin, isovitexin, kaempferol, ginsenoside Rg_1_, luteolin, taurocholic acid (TCA), ginsenoside Rb_1_, notoginsenoside Fa, beta-Ionone, taurohyodeoxycholic acid (THDCA), taurochenodeoxycholic acid (TCDCA), soyasaponin I, ginsenoside Rh_4_, chenodeoxycholic acid (CDCA) and betulonic acid, have significant effects on protecting the liver and gallbladder, and they are material basis of the efficacy for JGCC in the treatment of DHJS, as shown in Table 1.

### 2.3. Metabolite Analysis of JGCC into Blood

In the following, the metabolite M_21_ obtained from ginsenoside Rg_1_ was used as an example to describe the process of characterizing metabolites for JGCC that can be found in the blood. Under the positive ion mode, the parent ion peak *m*/*z* 855.48 of the metabolite M_21_ was extracted at t_R_ = 27.51 min (Figure 6A) and it was the oxidation product of ginsenoside Rg_1_. The parent ion *m*/*z* 855.48 and the fragment ions *m*/*z* 546.36, 487.29 and 323.26 were scanned in the high and low energy channels (the high collision energy was 20–40 V and the low collision energy was 6 V) (Figure 6C,D). Through analysis of the unsaturation and elemental composition, it was found that an *m*/*z* 546.36 was obtained by dropping C_10_H_22_O_9_ from M_21_, an *m*/*z* 487.29 was obtained by dropping C_17_H_29_O_7_ from M_21_ and an *m*/*z* 323.26 was the result of dropping C_22_H_37_O_13_ from M_21_ (Figure 7).

A total of 33 metabolites were characterized and identified (Table 2), and they were derived from 12 prototype components, including abrine and afrormosin from *Abrus cantoniensis* Hance, scoparone and capillarisin from *Artemisia capillaris* Thunb, geniposide from *Gardenia jasminoides* Ellis, ginsenoside Rb_1_, ginsenoside Rg_1_, ginsenoside Rd and notoginsenoside T_5_ from *Panax notoginseng* (Burk.) F.H.Chen, cholic acid from Bovis calculus Artifactus, CDCA from *Sus scrofadomestica* Brisson, and the common component of hyodeoxycholic acid (HDCA) from Bovis calculus Artifactus *and Sus scrofadomestica* Brisson. After consulting the literature, it is found that nine components, including abrine, scoparone, capillarisin, geniposide, ginsenoside Rb_1_, ginsenoside Rg_1_, ginsenoside Rd, CDCA and HDCA, have significant activities of protecting the liver. They are also the material basis for the efficacy of JGCC in treating DHJS.

### 2.4. Determination of Q-Markers for JGCC

According to the determination principles of Q-markers of TCM proposed by academician Changxiao Liu, we found the Q-markers of JGCC. The specific process is as follows.

(1)The pharmacodynamic components of JGCC. Through the literature review, it was found that 27 of the 43 prototype blood components were successfully characterized and identified, and they have obvious hepatoprotective effects. The nine prototype components of metabolites have obvious protective effects on the liver. Finally, trigonelline, 3,4,5-trihydroxybenzoic acid, geniposide acid, abrine, chlorogenic acid, hypaphorine, *p*-coumaric acid, geniposide, genipin-1-gentiobioside, vicenin-2, albiflorin, isoschaftoside, paeoniflorin, isovitexin, kaempferol, ginsenoside Rg_1_, luteolin, TCA, ginsenoside Rb_1_, notoginsenoside Fa, beta-ionone, THDCA, TCDCA, soyasaponin I, ginsenoside Rh_4_, CDCA, betulonic acid, scoparone, capillarisin, ginsenoside Rd and HDCA are considered to be the pharmacodynamic material basis of JGCC in the treatment of DHJS.(2)The inherent components of JGCC. The inherent components mean the prototype components of the pharmacodynamic material basis in JGCC, so the 31 components mentioned in (1) are also the inherent components of JGCC.(3)The unique ingredients in the herbal medicine of JGCC. Among the inherent pharmacodynamic components, chlorogenic acid is a common component of *Artemisia capillaris* Thunb and *Gardenia jasminoides* Ellis. Kaempferol is a common component of *Paeonia lactiflora* Pall, *Origanum vulgare* L. and *Gardenia jasminoides* Ellis. HDCA is a common component of *Sus scrofadomestica* Brisson and Bovis calculus Artifactus. The remaining components are the unique ingredients of each herb.(4)The measurable components in JGCC. Yan [25] and Liu [26] established a method for the content determination of trigonelline, abrine, hypaphorine, isoschaftoside, isovitexin, luteolin and vicenin-2 in *Abri Herba* and *Abri Mollis Herba* by HPLC-MS/MS. The “Chinese Pharmacopoeia” includes the content determination methods for chlorogenic acid and scoparone in *Artemisia capillaris* Thunb, ginsenoside Rg_1_ and ginsenoside Rb_1_ in *Panax notoginseng* (Burk.) F.H.Chen, THDCA in *Sus scrofadomestica* Brisson and paeoniflorin in *Paeonia lactiflora* Pall. In addition, according to the domestic and foreign literature, the following components have been determined: geniposide, geniposidic acid and genipin-1-gentiobioside are present in *Gardenia jasminoides* Ellis [27], albiflorin is discovered in *Paeonia lactiflora* Pall [28], TCA and TCDCA are found in snake bile [29], CDCA is discovered in bio-transformed samples [30] and scoparone and capillarisin are found in *Artemisia capillaris* Thunb and its compound preparations [31]. These have all been previously quantified.(5)The prescription compatibility properties of Q-markers in JGCC. In JGCC, *Abrus cantoniensis* Hance is the monarch (jun), *Artemisia capillaris* Thunb and *Gardenia jasminoides* Ellis are the ministers (chen) and *Paeonia lactiflora* Pall, *Panax notoginseng* (Burk.) F.H.Chen, *Origanum vulgare* L., *Sus scrofadomestica* Brisson and Bovis calculus Artifactus are the adjuvants (zuo). *Lycium barbarum* L. and *Ziziphus jujuba* Mill are the ambassadors (shi). Therefore, the selection of Q-markers is mainly based on the specific pharmacodynamic components in the monarch medicine *Abrus cantoniensis* Hance, such as trigonelline, abrine, hypaphorine, isoschaftoside, isovitexin, luteolin, vicenin-2 and soyasaponin I, as well as the specific pharmacodynamic components of other medicinal materials.

Based on the pharmacodynamic material basis and Q-marker screening principles of JGCC, trigonelline, abrine, vicenin-2, hypaphorine, isoschaftoside, isovitexin, soyasaponin I, luteolin, scoparone, capillarisin, paeoniflorin, albiflorin, geniposide, geniposidic acid, genipin-1-gentiobioside, ginsenoside Rg_1_, ginsenoside Rb_1_, ginsenoside Rh_4_, ginsenoside Rd, notoginsenoside Fa, TCA, THDCA, CDCA, TCDCA, 3,4,5-trihydroxybenzoic acid, *p*-coumaric acid and betulonic acid can be used as the Q-markers of JGCC.

### 2.5. Specificity, Linear Range, Limit of Detection (LOD), Limit of Quantification (LOQ)

We selected the Q-markers with higher content in JGCC and established a simultaneous determination method for the multiple components, aiming to lay the foundation for the improvement of the quality standard of JGCC. A 70% methanol solvent had no effect on the quantitative determination of the 16 active ingredients in JGCC. The results of the linear regression showed that all the compounds had good linear correlation in the concentration range, and the correlation coefficients were R^2^ ≥ 0.9993. The LOD is the lowest amount of a compound that can be detected in a sample. The LOQ is the lowest amount of a compound in a sample that can be quantitatively determined. The LOD of the 16 components were all greater than or equal to 0.07 ng/mL, and LOQs were all greater than or equal to 0.36 ng/mL. The regression equation, linear range, LOD and LOQ of the tested components are shown in Table 3.

### 2.6. Precision, Stability and Accuracy

The RSD of each component in the repeatability experiment was less than or equal to 8.82%, and the RSD of the inter-day precision was less than or equal to 9.23%. The JGCC solution and the mixed standard solutions were stable within 72 h, with RSD less than or equal to 9.64 and 9.81%, respectively. The recoveries of the 16 components ranged from 93.15% to 108.92%, and the RSD values were all less than 9.48%. This shows that the developed method was accurate, stable and reliable. The results are shown in Appendix A.

### 2.7. Determination Results of Multiple Batches of JGCC

The contents of the 16 components in 14 batches of JGCC were determined using the explained method, and the results are shown in Table 4. The average content of compounds **1**–**16** was 0.06–5.40 mg/g.

## 3. Discussion

*Abrus cantoniensis* Hance is the monarch medicine in JGCC, and it is mainly distributed in the Guangdong and Guangxi regions of China. It has the effect of dislodging dampness and jaundice, clearing heat and detoxification, soothing the liver and relieving pain, and is commonly used for DHJS, flank discomfort, epigastric distension and breast carbuncle swelling pain [25]. It can also be used to make soup for food therapy in the wet seasons such as the spring and summer [32]. The main active ingredients of *Abrus cantoniensis* Hance were abrine, trigonelline, isoschaftoside, hypaphorine and vicenin-2 in this study. Experimental studies have shown that abrine can be used to treat liver cancer, where it can inhibit the growth of liver tumors both in vitro and in vivo. It also reduces the levels of PD-L_1_ and KAT_5_ and regulates the growth and apoptosis of liver cancer cells through the KAT_5_/PD-L_1_ axis, and regulates the growth of liver cancer cells and the proliferation and activation of T cells [33]. Trigonelline can block the impaired autophagy of hepatocytes induced by high cholesterol and high fat diets, which prevents steatosis, so as to treat nonalcoholic fatty liver disease (NAFLD) [34]. Isoschaftoside can significantly reduce lipid deposition in cells, reverse NAFLD and reduce hepatic steatosis in mice, and it has pharmacological effects such as liver protection, anti-inflammatory, anti-tumor, heat-clearing and dampness removal, which has the potential for clinical applications [35]. Abrine, hypaphorine and vicin-2 in *Abrus cantoniensis* Hance improve the biochemical blood indexes of laying hens with fatty liver hemorrhage syndrome by reducing AST and ALT, triglycerides, low-density lipoprotein cholesterol (LDL-C) and total cholesterol and increasing the levels of high-density lipoprotein cholesterol [36].

Ginsenoside Rg_1_, ginsenoside Rb_1_ and notoginsenoside Fa are all derived from *Panax notoginseng* (Burk.) F.H.Chen [24]. Ginsenoside Rg_1_ competitively inhibits 2,3,7,8-tetrachlorodibenzodioxin (TCDD)-induced cytochrome P450 1A1 mRNA transcription by regulating aryl hydrocarbon receptor nuclear translocation, and it is a potent AhR agonist, so it can become a potentially effective drug for the prevention of TCDD-related liver injury [37]. Ginsenoside Rg_1_ exerts an anti-apoptotic effect on nonalcoholic fatty liver cells [38]. Ginsenoside Rb_1_ has a certain protective effect on liver toxicity induced by cantharidin in SD rats, which may be related to the up-regulation members of the MAPK family such as *p*-ERK, *p*-JNK and *p*-p38MAPK [39]. Ginsenoside Rb_1_ also has a protective effect on immune liver injury induced by restriction stress combined with lipopolysaccharide in mice, and its mechanism may be related to the up-regulation of deacetylase sirtuin-3/forkhead box transcription factor O_3_/super oxide dismutase function [40].

Chlorogenic acid is a common component of *Gardenia jasminoides* Ellis and *Artemisia capillaris* Thunb [24]. Studies [41,42] have shown that it has significant anti-inflammatory, hepatoprotective, choleretic and antioxidant effects, and it can reduce liver fibrosis in mice with nonalcoholic steatohepatitis; this hepatoprotective effect is attributed to the regulation of gut–liver axis homeostasis. Geniposide has been shown to have hepatoprotective, choleretic and hypoglycemic effects, and it can also improve ethanol-induced apoptosis and treat alcoholic liver injury; the therapeutic mechanism is related to the citric acid cycle and energy metabolism [43,44]. Although our study did not find the prototypic blood components of scoparone and capillarisin, their metabolites showed a high response in the serum. Capillarisin has a protective effect on acute ethanol-induced liver injury in mice, and its mechanism is related to the enhancement of the ability of the liver to clear acetaldehyde and some antioxidants [45]. Scoparone can attenuate D-galactosamine/LPS-induced liver injury by inhibiting the Toll-like receptor (TLR)-mediated inflammatory pathway and reducing local inflammation through immunomodulation, and it can be used as an effective ingredient for treating Yang Huang syndrome [46]. Scoparone can significantly inhibit the proliferation and activation of hepatic stellate cells through the inactivation of the TGF-β/Smad signaling pathway, and it can also have an anti-hepatic fibrosis effect [47].

Furthermore, because the medicinal materials of JGCC contain *Sus scrofadomestica* Brisson and Bovis calculus Artifactus, many components are not only endogenous substances that exist in the blood, but also active ingredients contained in the products, such as CDCA and THDCA. In the process of identification, the response value of the administration group should be greater than or equal to two times that of the model group as the standard, so as to determine whether it was an exogenous component in the blood. CDCA is an agonist of the farnesol nuclear receptor and it can regulate the synthesis and transport of bile acids, so the reduction of CDCA can lead to the accumulation of high concentrations of bile and be harmful to the liver. It can also reduce the expression of the hepatic LDL-C receptor, thereby reducing the hepatic recycling of LDL-C, which can inhibit hepatocyte autophagy and intestinal cholesterol adsorption [48,49]. Carubbi et al. compared the cytotoxicity and cytoprotective effects of the hydrophilic bile acids, THDCA and tauroursodeoxycholic acid (TUDCA), on the HepG_2_ cell lines and they found that both of them had significant protective effects on HepG_2_ cells, but hemolysis occurred under prolonged exposure at high concentrations [50]. THDCA stimulates more cholesterol and phospholipid secretion than TUDCA with a higher phospholipid/cholesterol secretion ratio, and THDCA is not hepatotoxic [51].

The effectiveness of TCM is the core element of therapy and an important basis for the determination of Q-markers [52]. The so-called “active ingredients” mentioned and measured in the current quality standards of most TCMs and compound prescriptions are mostly the main ingredients of the source plants. The index ingredients are sometimes listed without sufficient evidence to prove that they are active ingredients. Even though most TCMs and components are active in vitro, they may have no obvious effect in vivo and cannot be absorbed, or they need to be metabolized in order to produce the active substances. Previous studies on JGCC either focused on each single medicinal herb or on the content determination of several simple components, without the support of the pharmacological substance basis [32,53,54]. In this study, the method of serum pharmacochemistry of TCM was used for the first time to clarify the pharmacodynamic material basis of JGCC, and then, according to the determination principle of Q-markers, trigonelline, abrine, geniposide, etc., were determined as Q-markers of JGCC. The innovation of this study is that most of the previous studies were administering drugs to healthy rats for blood component analysis, while this study is based on the successful modeling of DHJS and the effective treatment with JGCC, which is more practical. As is to all, there are differences in the composition of drugs entering the blood under health or disease conditions. The content determination index was selected according to Q-markers of JGCC. The established UPLC-MS/MS quality evaluation method for simultaneously quantifying multiple components was more scientific and reasonable than the previous determination of components, which provided a basis for the improvement of the quality standard in the future and ensured the effectiveness of the clinical application of JGCC.

## 4. Materials and Methods

### 4.1. Materials

Acquity UPLC, Synapt G2-Si Q-TOF-MS and the OASIS^®^ PRIME hydrophilic and lipophilic balance (HLB) solid-phase extraction cartridge (60 mg, 3 cc) were purchased from Waters, USA. The TSQ Quantis Plus triple quadrupole LC-MS was from Thermo Fisher, Waltham, MA, USA. The Hitachi 3100 automatic biochemical analyzer was from Nanning precision instrument Co., Ltd., Nanning, China. Chromatographic-grade methanol, acetonitrile and formic acid were purchased from Thermo Fisher, Waltham, MA, USA. Chromatography-grade phosphoric acid was from the Macklin Company. α-naphthalene isothiocyanate (ANIT) was from Aladdin and distilled water was supplied via the Guangzhou Watsons Food and Beverage Co., Ltd. (Guangzhou, China). Professor Xijun Wang of the Pharmacognosy Department, Heilongjiang University of Chinese Medicine, identified that *Zingiber officinale* Rosc (Guangxi Xianzhu Chinese Medicine Technology Co., Ltd. Nanning, China) was genuine. JGCC was provided by the Guangxi Yulin Pharmaceutical Group Co., Ltd. (Yulin, China), and all the specific detailed information relating to this product is shown in Appendix A. The standard substances abrine, THDCA and isovitexin were purchased from Shanghai Yuanye Biotechnology Co., Ltd. (Shanghai, China). The standard substances hypaphorine, ginsenoside Rg_1_, ginsenoside Rb_1_, notoginsenoside Fa, paeoniflorin, albiflorin, CDCA, trigonelline, geniposide, vicenin-2, genipin 1-gentiobioside, isoschaftoside and luteolin were all purchased from Sichuan Weikeqi Biotechnology Co., Ltd. (Chengdu, China). The purity of all reference substances was greater than or equal to 98% and their batch numbers are given in Appendix A.

### 4.2. Chromatographic Analysis

The chromatographic column was Waters ACQUITY UPLC^®^ HSS T_3_ (2.1 × 100 mm, 1.8 μm) and was eluted with a mixture of water (A, containing 0.1% formic acid) and acetonitrile (containing 0.1% formic acid). The elution program was as follows: 0–4 min, 98–80%A; 4–30 min, 80–30%A; 30–32 min, 30–0%A; 32–35 min, 0%A; 35–35.1 min, 0–98%A; and 35.1–37 min, 98%A. The column temperature was 40 °C. The column flow rate was 0.4 mL/min and the injection volume was 4 μL.

### 4.3. UPLC-Q-TOF-MS Conditions

Synapt G2-Si Q-TOF-MS was adopted as an electrospray ionization source (ESI) with high collision energy at 20–40 V and low collision energy at 6 V, and employed a full scan MS^E^ mode at *m*/*z* = 50–1500 Da. The optimized cone voltage was 40 V and the capillary voltage was 2200 V. The desolvent gas temperature was 400 °C with a flow rate of 800 L/h, and the ion source temperature was 105 °C. The cone gas flow rate was 50 L/h. Leucine-enkephalin ([M+H]^+^ = 556.2771, [M−H]^−^ = 554.2615) solvent was used as the locking mass solution for accurate mass determinations.

### 4.4. UPLC-QQQ-MS Conditions and Optimization

The TSQ Quantis Plus UPLC-QQQ-MS uses an H-ESI ion source and a mixed scan of positive and negative ions in the selected reaction monitoring (SRM) mode for quantitative analysis. A spray voltage with a positive ion of 3500 V and a negative ion of 2500 V was used. The ion transfer tube temperature was 325 °C and the evaporation temperature was 350 °C. The sheath gas was 40 Arb, the auxiliary gas was 10 Arb and the sweep gas was 0 Arb. The Q_1_ resolution was 0.7 and the Q_3_ resolution was 0.7. The collision gas pressure was 1.5 mTorr and the other optimized parameters used are shown in Appendix A.

### 4.5. Preparation of the Modeling Solution

Amounts of 130.0 mg and 90.0 mg of ANIT powder were weighed and 50 mL olive oil was added followed by ultrasonic dissolution, and 2.6 mg/mL and 1.8 mg/mL ANIT modeling solutions were prepared. An amount of 300 g of *Zingiber officinale* Rosc was added to 3 L distilled water and immersed for 1 h, heated to the boil and decocted for another 1 h with soft fire by an induction cooker. The mixture was filtered through four layers of gauze and 3 L of water was added to the residue; the above operation was repeated twice. The three filtrates were collected and concentrated to 3 L through decocting. The final *Zingiber officinale* Rosc modeling solution concentration was 0.01 g/mL after dilution.

### 4.6. Source and Processing of Serum Samples

The source of the serum samples was based on the samples of our previous metabolomics study on JGCC in the treatment of DHJS [6,7]. Eight-week-old clean-grade adult male SD rats were purchased from Guangxi Medical University (license number SYXK (Gui) 2020-0014). Rats were randomly divided into three groups of eight rats in each group. These was a control group, a model group and a JGCC group. At the beginning of the experiment, the model and JGCC groups were each administered with 0.7 mL/200 g of *Zingiber officinale* Rosc solution every morning and 12.5% ethanol solution at a dose of 1.0 mL/100 g every afternoon, which was repeated for 14 days. The ANIT olive oil solution was administered at doses of 10.4 mg/kg and 7 mg/kg on the 15th and 16th days, respectively, to prepare DHJS model rats. At the beginning of 17th day, 2.16 g/kg of JGCC was administered to the JGCC group, and the other groups were given an equal amount of water for 14 consecutive days. The rats were sacrificed on the 31st day after anesthesia, and fresh blood samples were collected through the abdominal aorta. The upper serum portion was taken and stored in a −80 ℃ refrigerator for later use. After collecting blood, the liver and bile duct of rats were immediately removed and the residual blood was washed with normal saline. They were fixed in 4% paraformaldehyde and analyzed by the HE staining method for histopathological evaluation. The animal study was approved by the Animal Care and Ethics Committee of Guangxi Botanical Garden of Medicinal Plants, approval number GXBGMP-20220201, 23 February 2022.

Two methods were used to prepare the serum. Method 1 was to take 1 mL of serum and add 1 mL of 4% phosphoric acid, and the mixture was agitated and centrifuged at 13,000 rpm for 10 min at 4 °C. The supernatant was added to the activated HLB solid-phase extraction column (which had been activated with 4 mL methanol and balanced with 2 mL water). The column was washed with 2 mL water and then with 1 mL 5% methanol and the eluent was discarded. Finally, it was washed with 2 mL of methanol and the eluate was collected. The methanol eluate was evaporated to dryness with a vacuum concentrator (40 °C), and the residue was washed with 200 μL methanol: acetonitrile 1:1. This was subjected to ultra-sonication and then it was centrifuged at 13,000 rpm for 10 min at 4 °C. The supernatant was taken for analysis. Method 2 was basically the same as method 1, except that the 4% phosphoric acid was not initially added.

### 4.7. ”Five Principles” for Determining Q-Markers

Q-markers of TCM can control the quality of medicinal materials and products, thus ensuring the effectiveness and safety of clinical drugs, so they are very significant. Q-markers of TCM must have the following five elements: (1) they can be the morphological characteristics, histological characteristics and genetic characteristics (such as DNA, DNA barcode) of TCM and decoction pieces, or the characteristic chemical substances and substance groups from Chinese medicinal materials, decoction pieces, extracts, single or compound preparations related to the quality for efficacy; (2) they are substances that can be qualitatively and quantitatively determined by chemical analysis and bioassay; (3) they have specificity of biological effects (effective and safe); (4) they have traceability of source and transmission of the industrial process; (5) under the guidance of the TCM theory, the rules of compatibility of prescriptions (such as the principle of taking “jun” medicine as the main medicine and “chen, zuo, shi” as assistants) should be reflected. In other words, Q-markers of TCM should be determined from five aspects: quality transfer and traceability, component specificity, component effectiveness, component testability and compound compatibility environment.

### 4.8. Preparation of JGCC for Quantitative Analysis

The preparation of the JGCC solution was same as the previously optimized method [24]. The specific steps were to weigh 0.1 g of JGCC and add 20 mL of 70% methanol, vibrate ultrasonically for 30 min and centrifuge at 4 °C for 10 min at 13,000 rpm, and the supernatant was used for instrument analysis.

### 4.9. Preparation of Mixed Standard Solutions

All the standard substances were dissolved in 100% methanol to prepare a 1 mg/mL single standard stock solution. These were then diluted with 70% methanol to form a series of mixed standard solutions. The serial concentrations of the 16 active ingredients are shown in Appendix A. By establishing the relationship between the peak area (Y) and concentration (X) of each component, standard curves were drawn.

### 4.10. Specificity, LOD, LOQ

To investigate the influence of solvents on the determination of each active component, 70% methanol was injected into the UPLC-QQQ-MS. On the basis of the linearity investigation, the mixed standard solutions were continuously diluted until the concentration with a signal/noise ratio of 3:1 was the LOD, and the concentration with a signal/noise ratio of 10:1 was the LOQ.

### 4.11. Precision

In order to evaluate the repeatability of the method, the content of JGCC (batch number 2111095) was prepared into 6 samples according to Section 4.8. The contents of the above 6 samples were measured on three consecutive days, and the RSD values were calculated to evaluate the inter-day precision of the method.

### 4.12. Stability

The mixed standard solution and sample 1 of the precision test were injected and analyzed at 0, 4, 8, 12, 24, 36, 48 and 72 h (stored in the sample chamber of the HPLC, dark, 10 ℃) to investigate the stability of the mixed standard solution and JGCC. The concentrations of compounds **1**–**16** in standard solution were 609.0 ng/mL, 499.0 ng/mL, 764.0 ng/mL, 511.0 ng/mL, 694.5 ng/mL, 694.5 ng/mL, 515.5 ng/mL, 575.5 ng/mL, 546.5 ng/mL, 538.0 ng/mL, 546.5 ng/mL, 558.0 ng/mL, 549.0 ng/mL, 525.5 ng/mL, 551.0 ng/mL and 525.0 ng/mL.

### 4.13. Accuracy

The recovery rate was tested by the standard addition method to evaluate the accuracy of the method. The contents of JGCC with the known concentration of each component were taken and nine samples were weighed. Three levels (low, medium and high) of standard solution were added at each level in three parallels. The formula used for calculating the recovery rate was [(measured amount-contained amount)/added amount] × 100%; the detailed data are shown in Appendix A.

### 4.14. Content Determination of the Multiple Samples

Using the established quantitative analysis method for the 16 compounds, the contents of multi-components in 14 batches of JGCC were determined simultaneously in order to evaluate their quality.

### 4.15. Data Analysis

The collected serum pharmacochemical MS^E^ data were analyzed by using the UNIFI data automatic processing software. According to the self-built database, the analysis conditions were set and the mass error was less than or equal to 5 mDa. The chromatographic peak extraction time was 0–30 min. In the positive mode, the adduct ions were [M+H]^+^ and [M+Na]^+^. In the negative mode, the adduct ions were [M-H]^−^ and [M+COOH]^−^. The endogenous metabolic modes of phases I and II were glucuronidation (+C_6_H_8_O_6_), glycosylation (+C_6_H_10_O_5_), hydrogenation (+H_2_), oxidation (+O), sulfation (+SO_3_), acetylation (+C_2_C_2_O), deglycosylation (−C_6_H_10_O_5_), desaturation (−H_2_) and deoxygenation (−O). The target ions were locked to the components that only existed in the JGCC or the response value was greater than or equal to 2 times that of the model group. According to the previously identified 144 active components of JGCC in vitro [24] and the retention time of ±0.5 min as the time window, the prototype components and metabolites of JGCC in the treatment of DHJS were analyzed and identified.

The quantitative analysis data of active components in JGCC were processed by using the TraceFinder software of the TSQ Quantis Plus UPLC-QQQ-MS. In each case, the ion with the highest peak intensity was selected as the quantitative ion. The experimental procedure is shown in Figure 8.

## 5. Conclusions

In this study, high-throughput mass spectrometry combined with the serum pharmacochemistry technology of TCM was adopted to clarify the pharmacodynamic material basis and discover the Q-markers of JGCC for the first time, and these Q-markers can be used as the quality control index of JGCC. The established UPLC-MS/MS method for simultaneous quantitative analysis of multiple components has high sensitivity, stability and accuracy, which can be used for the quality control of JGCC. This research can be referred to in the future in-depth study of the quality standard of JGCC. On the premise of understanding the content composition of each component, the components with high content and good ultraviolet absorption can be selected as the control index to develop the content determination method by HPLC, so as to meet the quality control requirements of JGCC in the actual production process. On the other hand, the production technology can also be improved according to the content of various active ingredients to achieve the purpose of effective dosage reduction. The study makes up the blank in the research of pharmacodynamic substances’ basis and provides a supplement for the quality standard of JGCC.

## Figures and Tables

**Figure 1 molecules-28-02494-f001:**
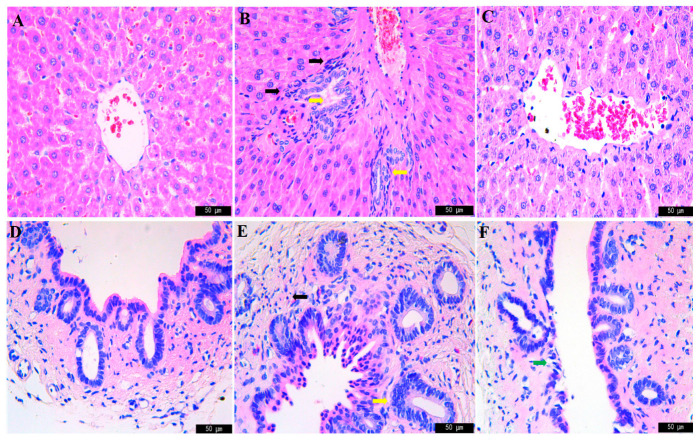
HE staining results of liver and bile duct with microscope (×400). (**A**–**C**): liver tissue, (**D**–**F**): bile duct tissue, (**A**,**D**): control group, (**B**,**E**): model group, (**C**,**F**): JGCC group. Black arrow: inflammatory cells, yellow arrow: hyperplasia of bile duct and bile duct epithelium, green arrow: degeneration and necrosis of bile duct epithelial cells.

**Figure 2 molecules-28-02494-f002:**
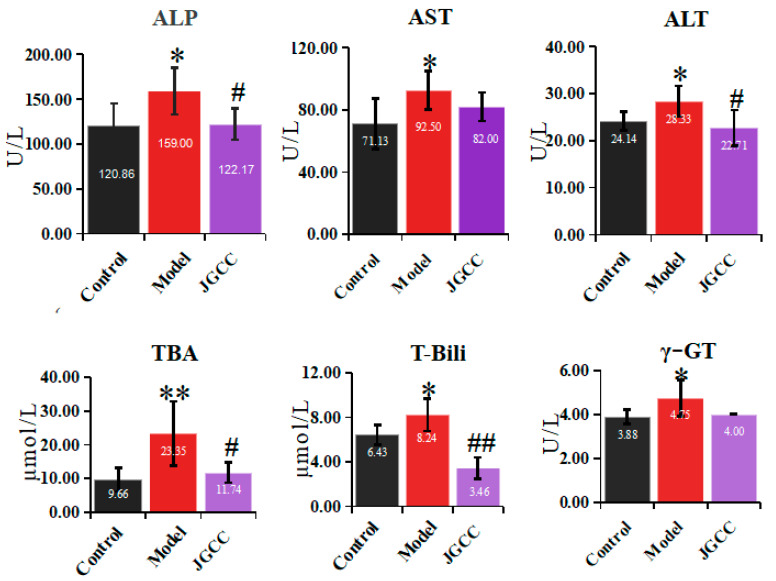
Determination results of biochemical indexes in different group vs. control group, * *p* < 0.05, ** *p* < 0.01; vs. model group, # *p* < 0.05, ## *p* < 0.01.

**Figure 3 molecules-28-02494-f003:**
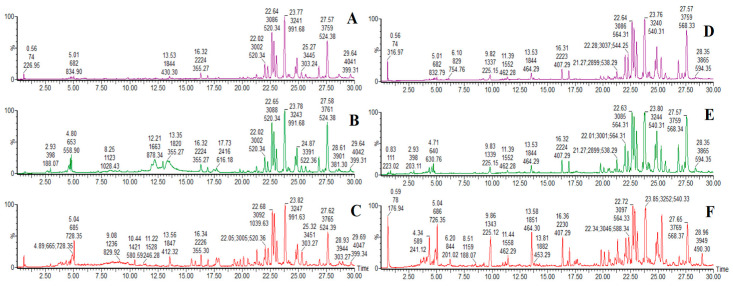
BPI chromatograms of rat serum samples. (**A**,**B**,**D**,**E**): serum samples of DHJS rats after 14 days of treatment with JGCC; (**C**,**F**): serum samples of DHJS rats; (**A**,**D**): method 2 to prepare serum samples; (**B**,**C**,**E**,**F**): method 1 to prepare serum samples; (**A**–**C**): positive ion mode; (**D**–**F**): negative ion mode.

**Figure 4 molecules-28-02494-f004:**
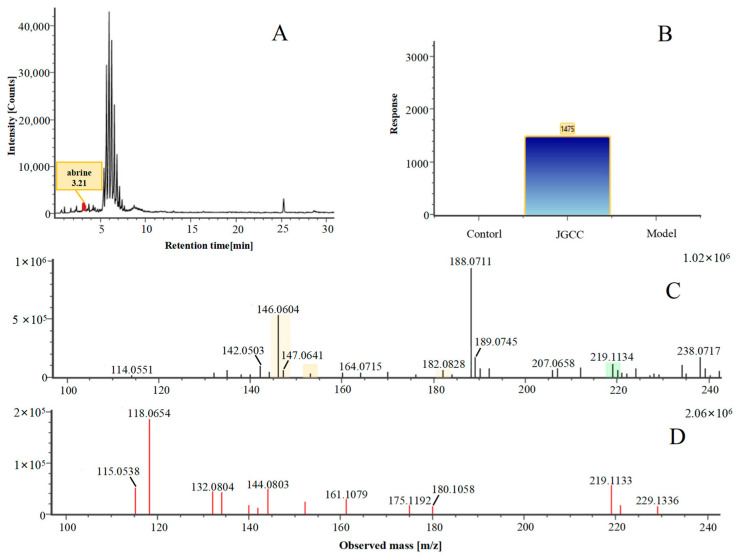
The characterization and identification process of abrine, the prototype component of JGCC found in the blood, based on the UNIFI data processing platform. (**A**): extraction of chromatographic peaks of abrine. (**B**): comparison of response values of abrine in control group, model group and JGCC group after the ingredients entered into the blood. (**C**): MS/MS information of abrine under low-energy collision; (**D**): MS/MS information of abrine under high-energy collision.

**Figure 5 molecules-28-02494-f005:**
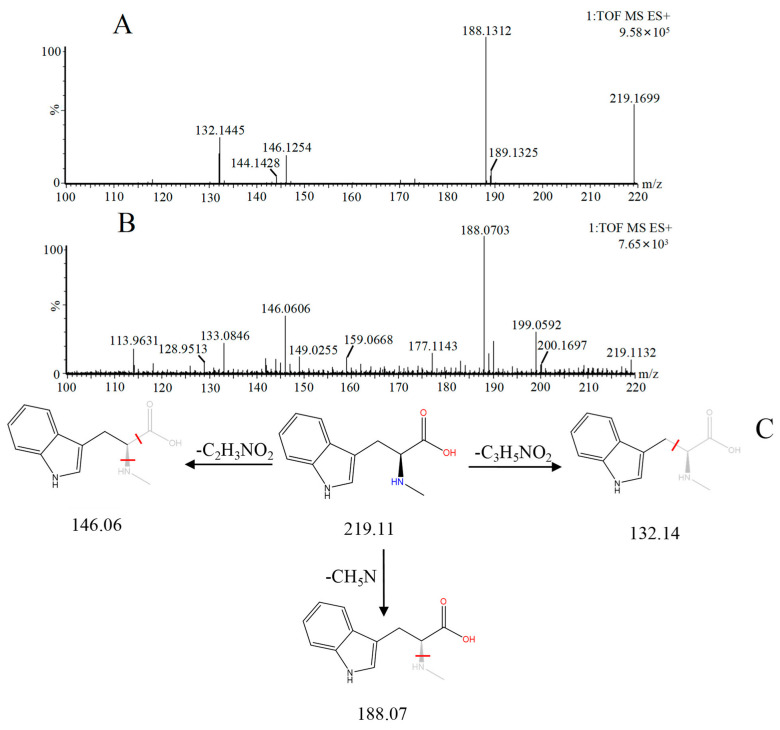
MS/MS fragment information and cleavage pathway of abrine. (**A**): MS/MS information of the abrine standard, (**B**): MS/MS information of abrine in serum samples, (**C**): the cleavage pathway of abrine.

**Figure 6 molecules-28-02494-f006:**
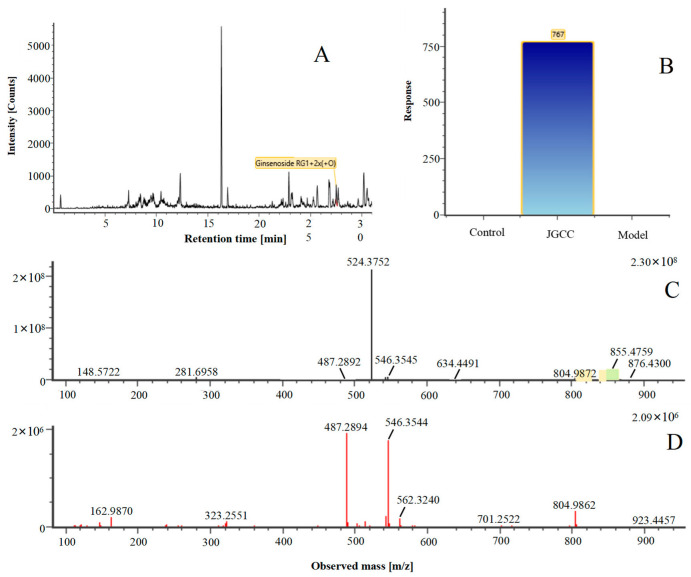
The characterization and identification of the blood component metabolite M_21_ in JGCC based on the UNIFI data processing platform. (**A**): chromatographic peak extraction diagram of the metabolite M_21_ from Ginsenoside Rg_1_, (**B**): comparison of response values in serum of M_21_ for control, model and JGCC group, (**C**): MS/MS information of M_21_ under low-energy collision, (**D**): MS/MS information of M_21_ under high-energy collision.

**Figure 7 molecules-28-02494-f007:**
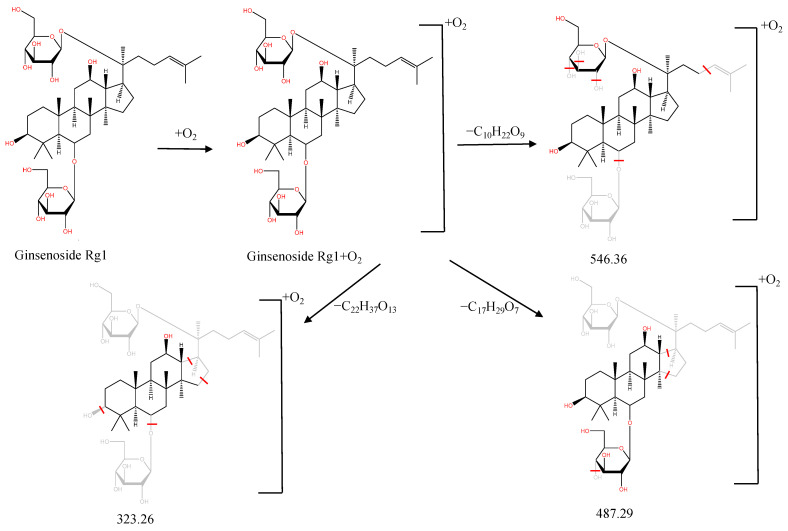
MS/MS information and the cleavage pathway of metabolite M_21_ from ginsenoside Rg_1_.

**Figure 8 molecules-28-02494-f008:**
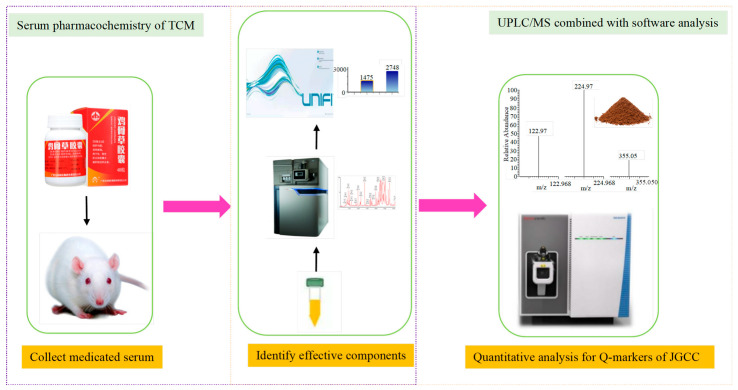
The technical flow chart for the identification of the different components in the blood after ingestion of JGCC and the establishment of the quantitative analysis method in vitro.

**Table 1 molecules-28-02494-t001:** The prototype blood components of JGCC.

NO	Compound	Rt	Observed *m*/*z*	Molecular Formula	MS/MS	References	Structural Formula
1	Trigonelline *^,#^	0.66	138.10	C_7_H_7_NO_2_	138.10[M+H]^+^,120.13[M+H−H_2_O]^+^	*Abrus cantoniensis* Hance	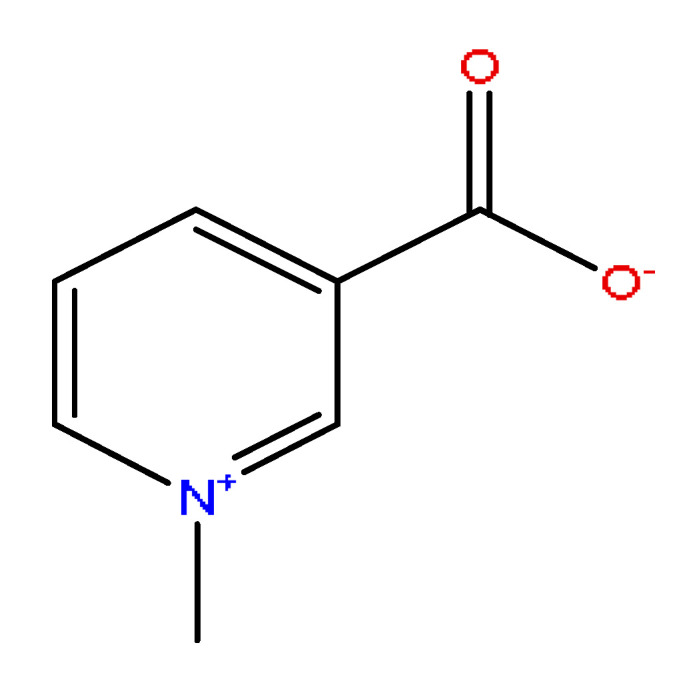
2	Citric acid	1.12	215.02	C_6_H_8_O_7_	215.012[M+Na]^+^,166.08[M+Na−CH_5_O_2_]^+^, 149.02[M+Na−CH_6_O_3_]^+^	*Lycium barbarum* L.	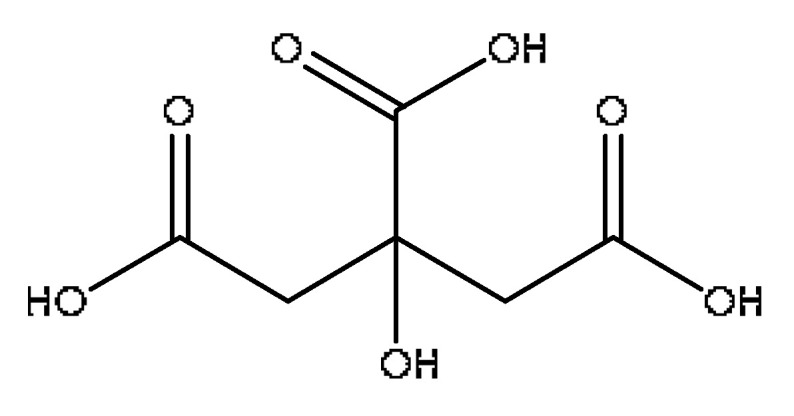
3	3,4,5-trihydroxybenzoic acid ^#^	1.80	169.02	C_7_H_6_O_5_	169.02[M−H]^−^,141.87[M-H−CO]^−^, 125.02[M-H−CO2]^−^	*Paeonia lactiflora* Pall	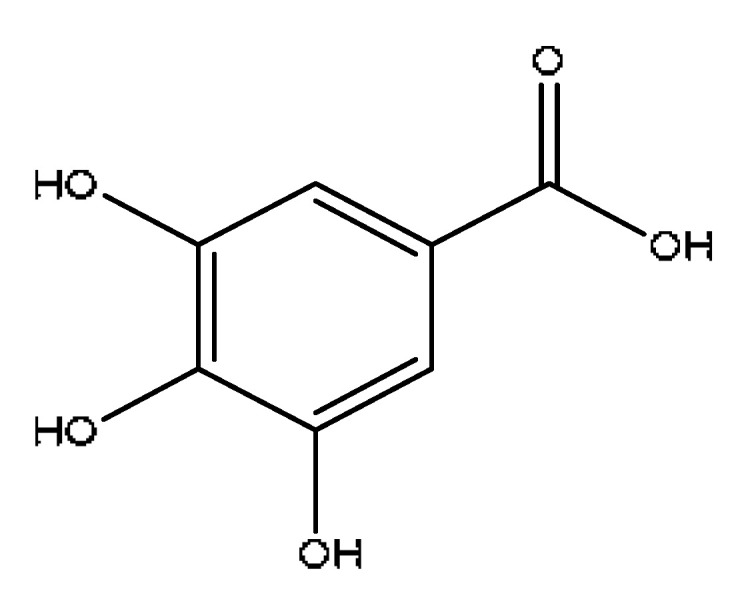
4	Progallin A	2.38	197.05	C_9_H_10_O_5_	197.05[M−H]^−^,180.91[M−H−OH]^−^, 168.03[M-H−C_2_H_5_]^−^,124.04[M-H-C_3_H_5_O_2_]^−^	*Paeonia lactiflora* Pall	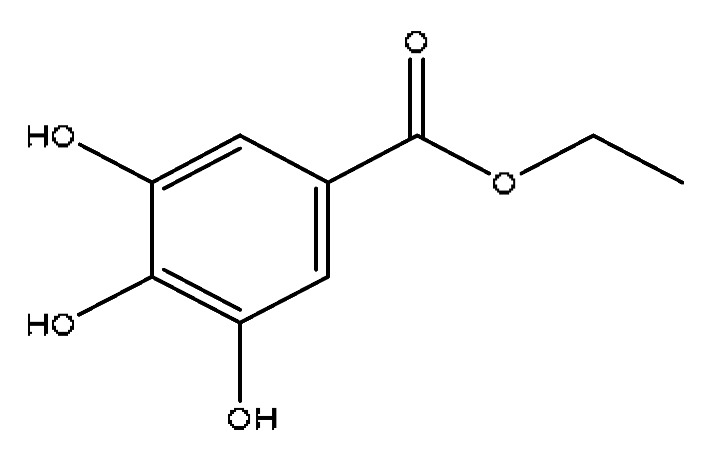
5	7-Methoxycoumarin	2.46	177.05	C_10_H_8_O_3_	177.05[M+H]^+^,159.09[M+H−H_2_O]^+^, 149.06[M+H−C_2_H_4_]^+^,146.06[M+H−CH_3_O]^+^	*Artemisia capillaris* Thunb	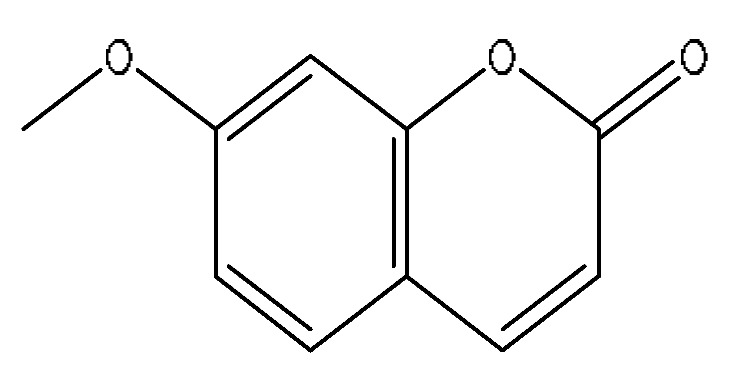
6	Scandoside methyl ester	2.65	449.13	C_17_H_24_O_11_	449.13[M+FA−H]^−^,354.11[M+FA−H−C_2_H_7_O_4_]^−^	*Gardenia jasminoides* Ellis	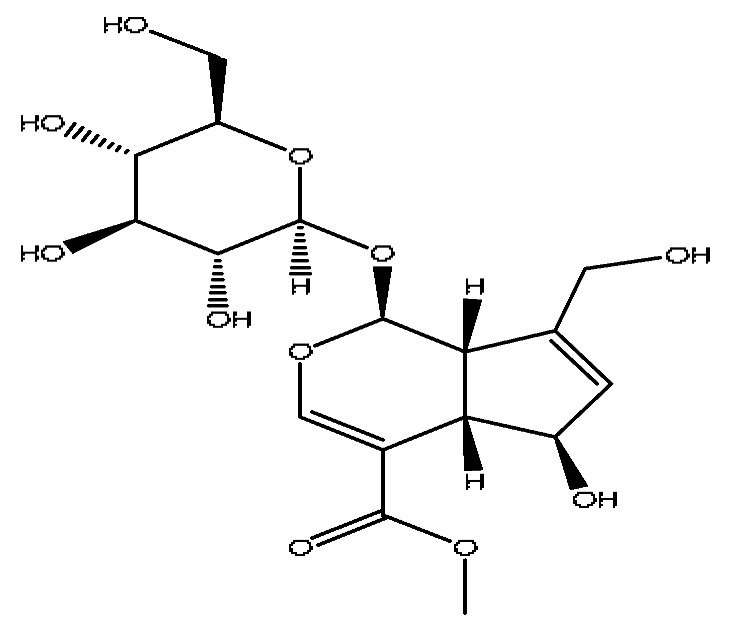
7	Geniposidic acid *^,#^	2.89	373.11	C_16_H_22_O_10_	373.11[M−H]^−^, 271.07[M−H−C_4_H_6_O_3_]^−^	*Gardenia jasminoides* Ellis	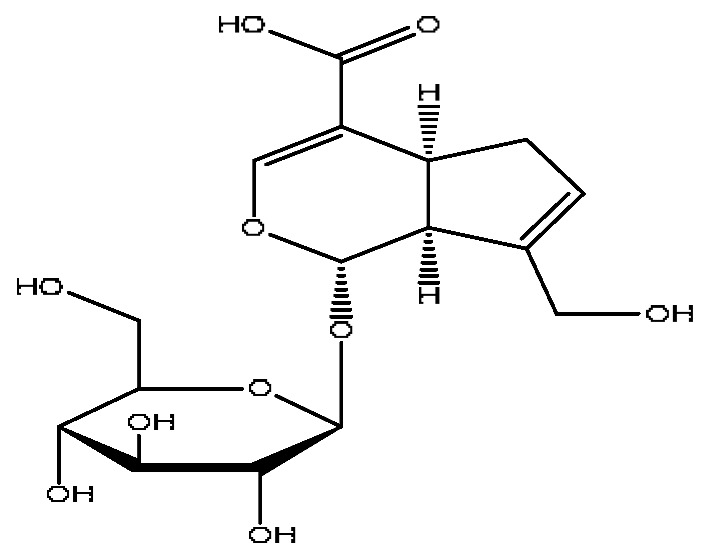
8	Scopolin	3.16	353.09	C_16_H_18_O_9_	353.08[M−H]^−^,228.09[M−H−C_6_H_5_O_3_]^−^, 205.07[M−H−C_5_H_8_O_5_]^−^	*Lycium barbarum* L.	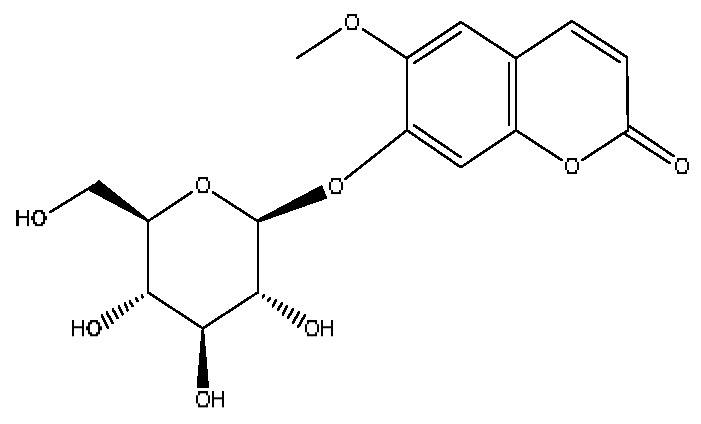
9	Abrine *^,#^	3.21	219.11	C_12_H_14_N_2_O_2_	219.11[M+H]^+^,188.07[M+H−CH_5_N]^+^, 146.06[M+H−C_2_H_3_NO_2_]^+^	*Abrus cantoniensis* Hance	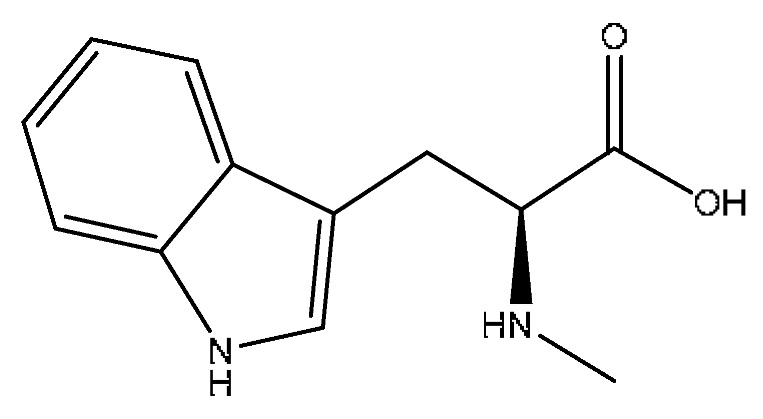
10	Chlorogenic acid *^,#^	3.40	353.08	C_16_H_18_O_9_	353.08[M−H]^−^,336.07[M−H−OH]^−^, 212.00[M−H−C_3_H_9_O_6_]^−^	*Gardenia jasminoides* Ellis, *Artemisia capillaris* Thunb	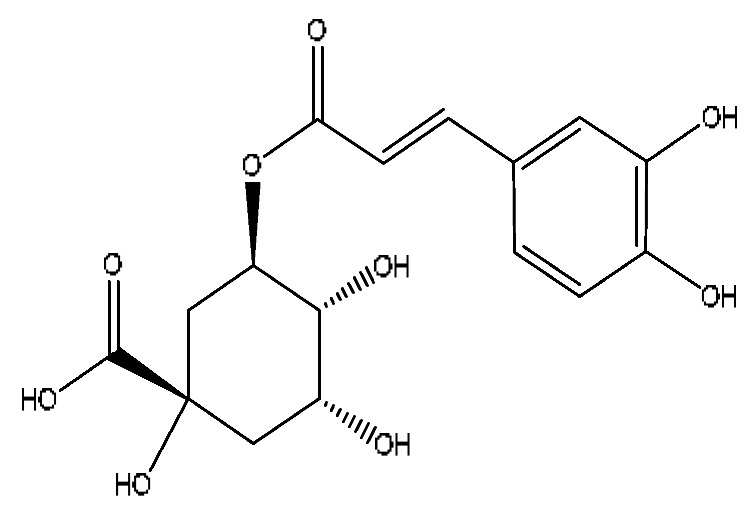
11	Hypaphorine *^,#^	3.42	247.18	C_14_H_18_N_2_O_2_	247.18[M+H]^+^,188.1142[M+H−C_3_H_9_N]^+^, 144.1232[M+H−C_4_H_9_NO_2_]^+^	*Abrus cantoniensis* Hance	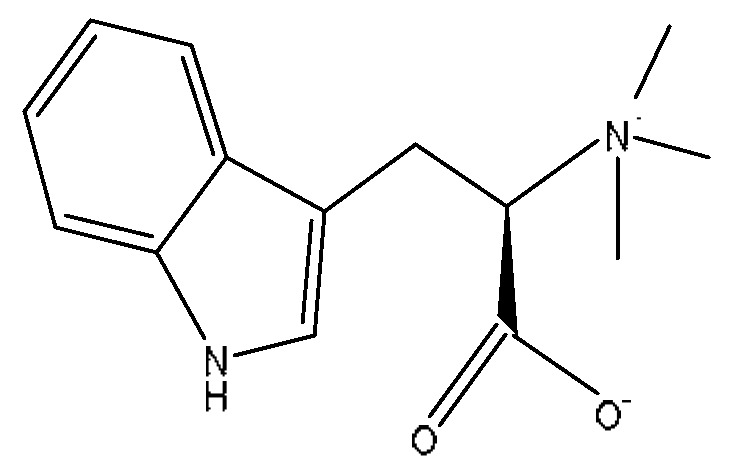
12	*p*-coumaric acid *^,#^	3.51	163.04	C_9_H_8_O_3_	163.04[M−H]^−^,146.96[M−H−OH]^−^, 119.05[M−H−CO_2_]^−^	*Ziziphus jujuba* Mill	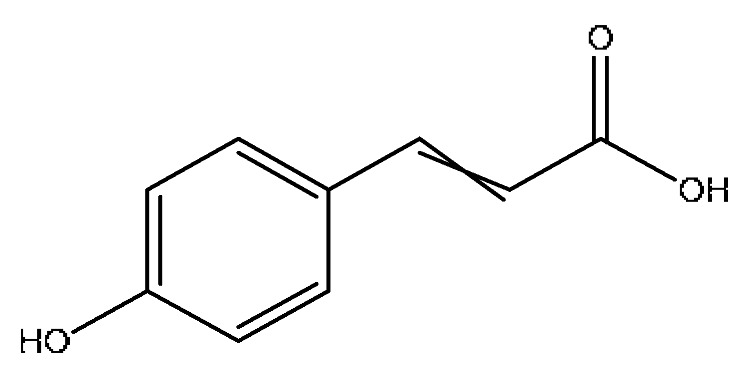
13	Geniposide *^,#^	3.56	387.13	C_17_H_24_O_10_	387.13[M−H]^−^,353.08[M−H−H_2_O_2_]^−^, 212.00[M−H−C_7_H_11_O_5_]^−^	*Gardenia jasminoides* Ellis	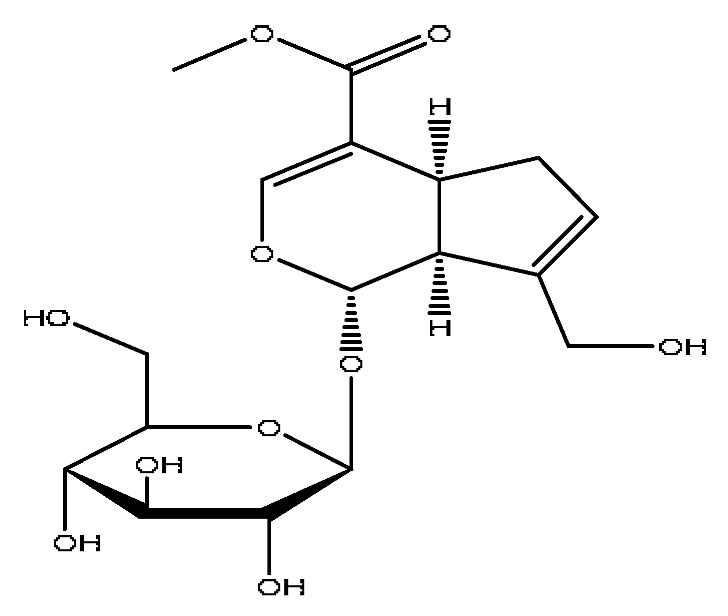
14	Ethyl caffeate *	4.01	209.08	C_11_H_12_O_4_	209.08[M+H]^+^,191.07[M+H−H_2_O]^+^, 177.11[M+H−CH_4_O]^+^	*Origanum vulgare* L.	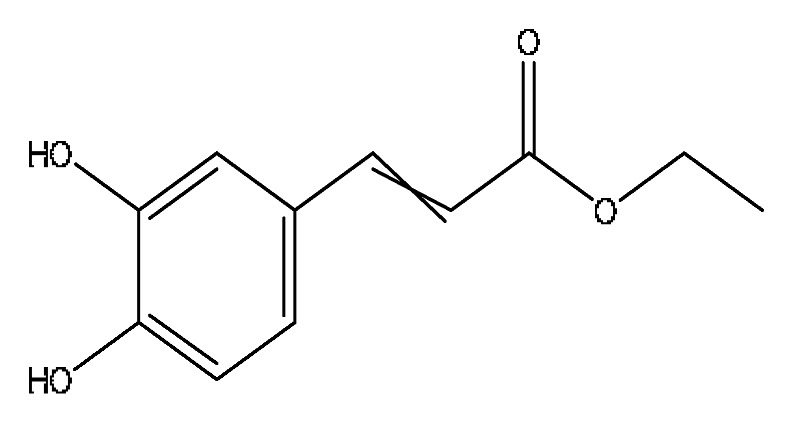
15	Safrol	4.03	207.07	C_10_H_10_O_2_	207.07[M+FA−H]^−^,180.91[M+FA−H−C_2_H_3_]^−^, 168.10[M+FA−H−C_3_H_3_]^−^, 153.02[M+FA−H−C_4_H_6_]^−^	*Lycium barbarum* L.	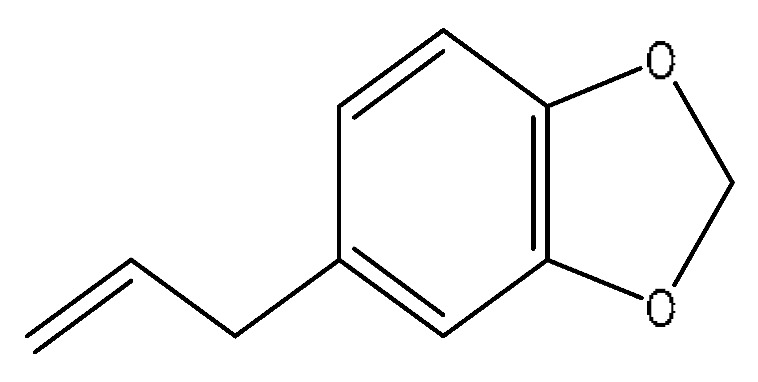
16	Genipin-1-gentiobioside *^,#^	4.03	549.15	C_23_H_34_O_15_	549.15[M−H]^−^,533.19[M−H−O]^−^, 505.22[M−H−CO_2_]^−^,255.10[M−H−C_8_H_22_O_11_]^−^	*Gardenia jasminoides* Ellis	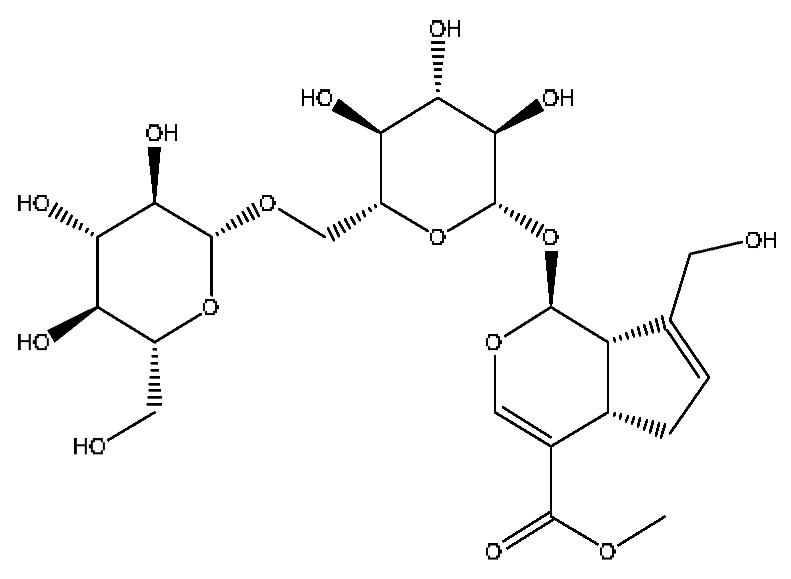
17	Vicenin-2 *^,#^	4.05	595.16	C_27_H_30_O_15_	595.16[M+H]^+^,523.22[M+H−H_8_O_4_]^+^	*Abrus cantoniensis* Hance	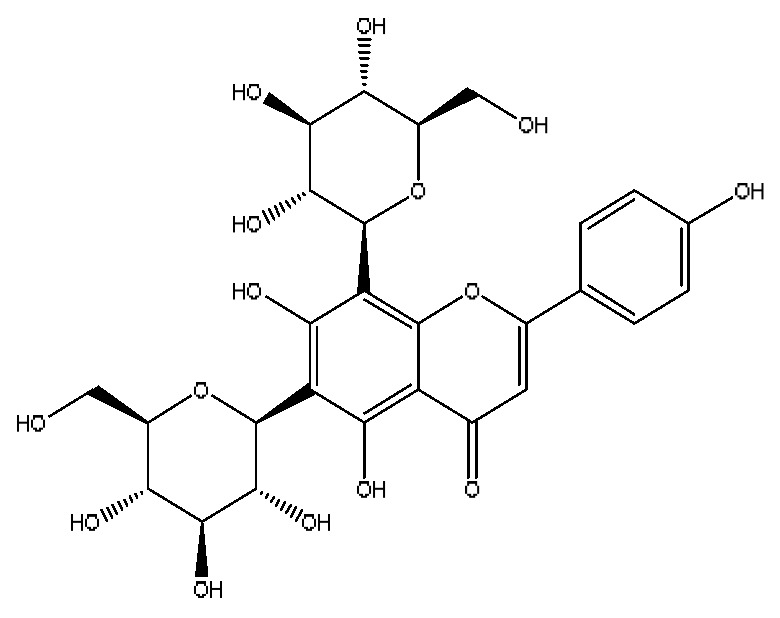
18	Albiflorin *^,#^	4.32	503.15	C_23_H_28_O_11_	503.15[M+Na]^+^,472.28[M+Na−CH_3_O]^+^, 455.26[M+Na−CH_4_O_2_]^+^,437.24[M+Na−CH_6_O_3_]^+^	*Paeonia lactiflora* Pall	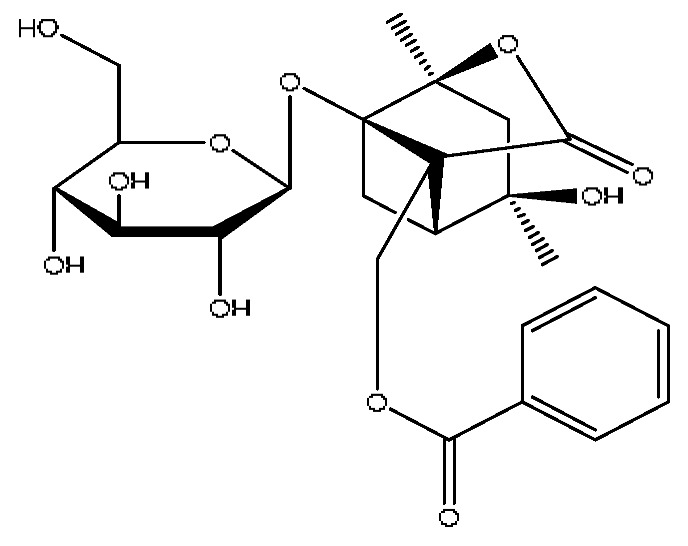
19	Isoschaftoside *^,#^	4.51	563.14	C_26_H_28_O_14_	563.14[M−H]^−^,427.23[M−H−C_8_H_8_O_2_]^−^, 283.08[M−H−C_11_H_20_O_8_]^−^	*Abrus cantoniensis* Hance	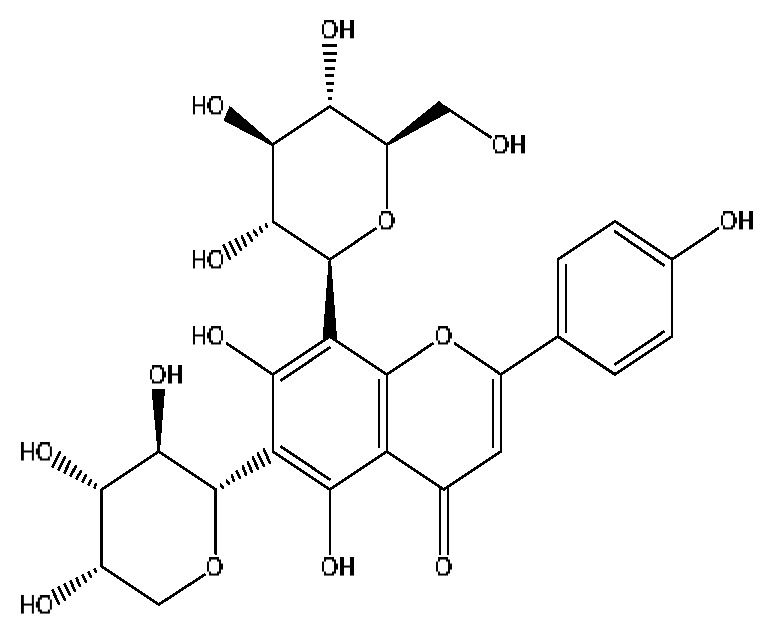
20	Paeoniflorin *^,#^	4.59	525.16	C_23_H_28_O_11_	525.16[M+FA−H]^−^,447.21[M+FA−H−C_6_H_6_]^−^, 283.08[M+FA−H−C_11_H_14_O_6_]^−^	*Paeonia lactiflora* Pall	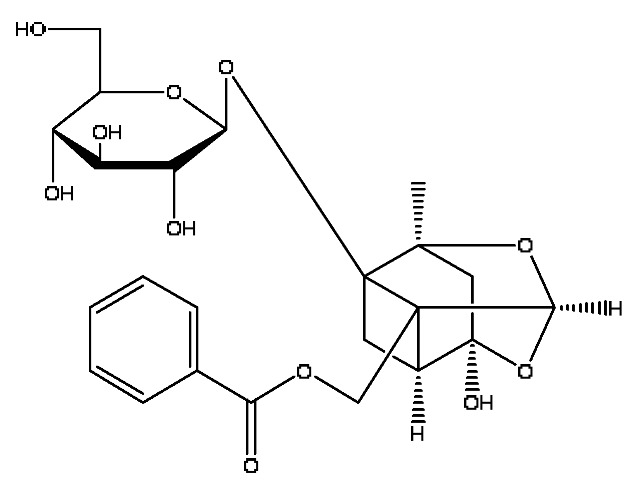
21	Isovitexin *^,#^	5.45	433.27	C_21_H_20_O_10_	433.27[M+H]^+^,414.27[M+H−H_3_O]^+^, 396.23[M+H−H_5_O_2_]^+^	*Abrus cantoniensis* Hance	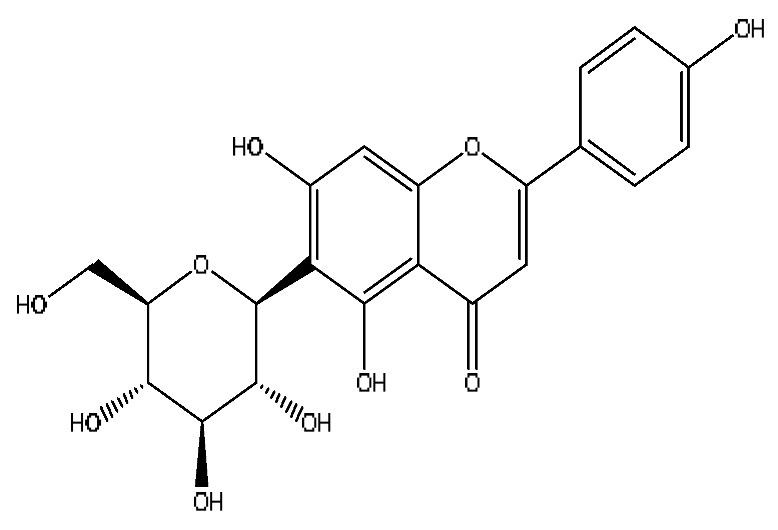
22	Kaempferol *^,#^	5.74	573.10	C_15_H_10_O_6_	573.10[2M+H]^+^,414.27[2M+H−C_7_H_11_O_4_]^+^, 382.22[2M+H−C_7_H_11_O_6_]^+^	*Paeonia lactiflora* Pall, *Gardenia jasminoides* Ellis, *Origanum vulgare* L.	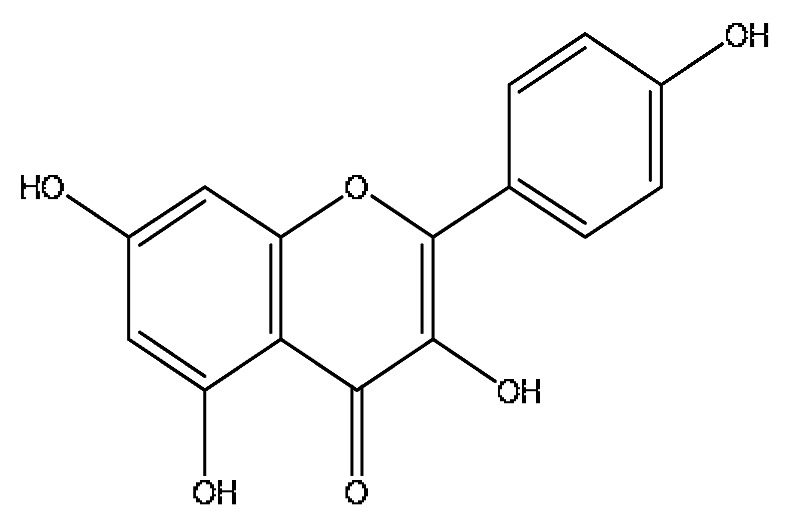
23	Ginsenoside Rg_1_ *^,#^	6.27	799.47	C_42_H_72_O_14_	799.40[M−H]^−^,767.43[M−H−O_2_]^−^, 417.12[M−H−C_14_H_22_O_12_]^−^	*Panax notoginseng* (Burk.) F.H.Chen	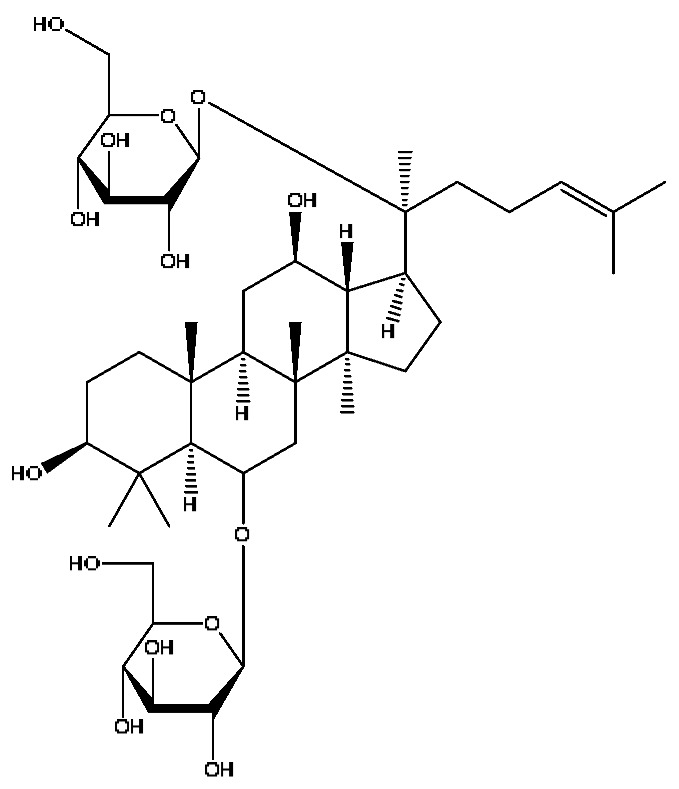
24	Mauritine A	7.45	576.32	C_32_H_41_N_5_O_5_	576.31[M+H]^+^,306.30[M+H−C_16_H_16_NO_3_]^+^, 262.27[M+H−C_18_H_22_N_2_O_3_]^+^	*Ziziphus jujuba* Mill	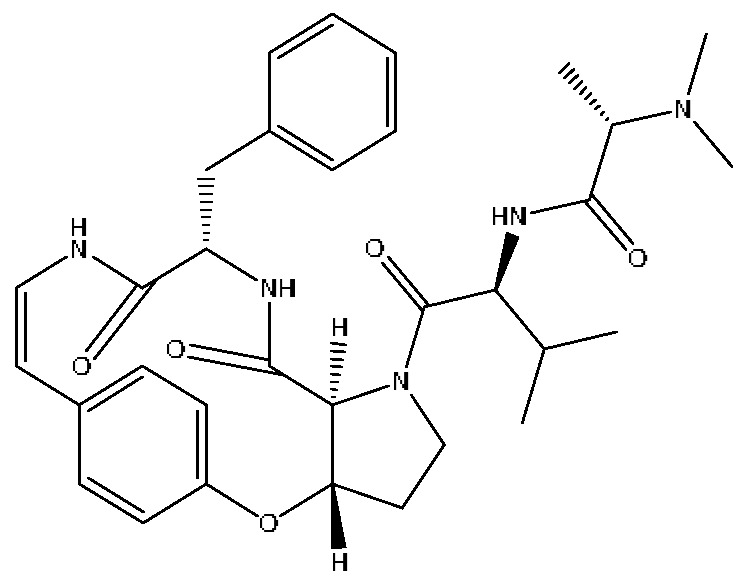
25	3′,4′,7-trihydroxyflavone	7.84	541.11	C_15_H_10_O_5_	541.11[2M+H]^+^,188.11[2M+H−C_19_H_13_O_7_]^+^, 170.10[2M+H−C_19_H_15_O_8_]^+^	*Abrus cantoniensis* Hance	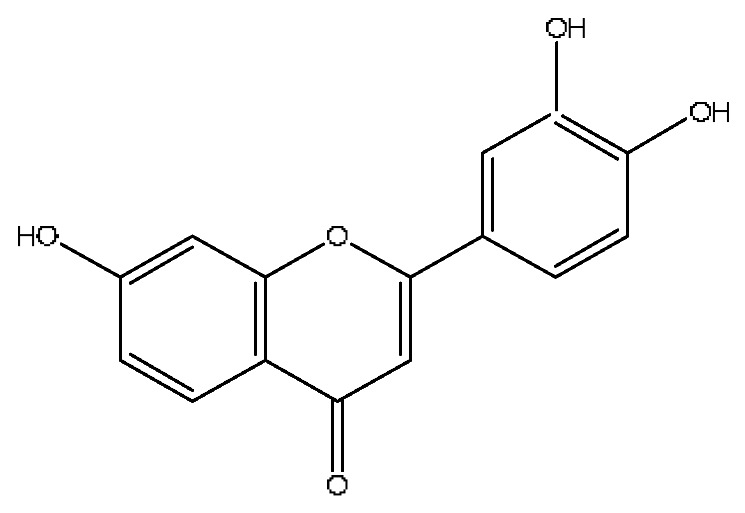
26	Luteolin *^,#^	8.61	285.04	C_15_H_10_O_6_	285.04[M−H]^−^,174.96[M−H−C_6_H_7_O_2_]^−^	*Abrus cantoniensis* Hance	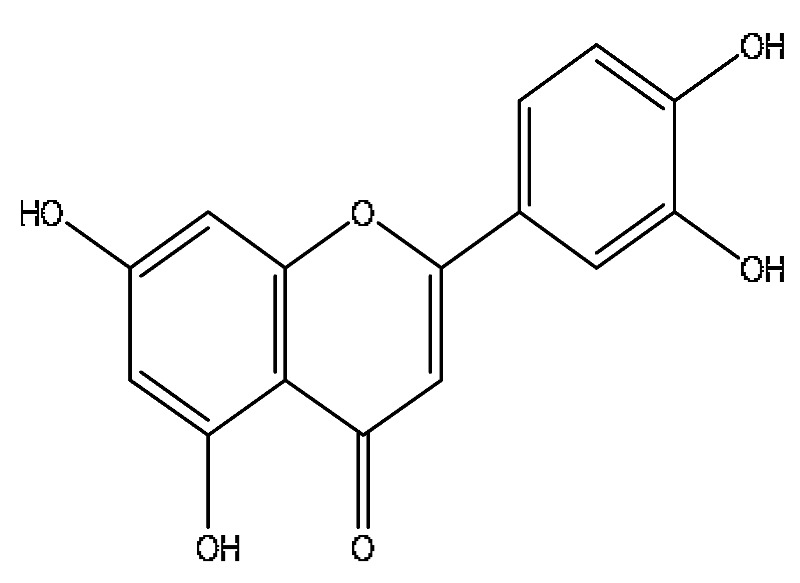
27	Apigenin 7-glucoside	8.62	477.11	C_21_H_20_O_10_	477.11[M+FA−H]^−^,461.10[M+FA−H−O]^−^, 188.07[M+FA−H−C_12_H_17_O_8_]^−^	*Origanum vulgare* L.	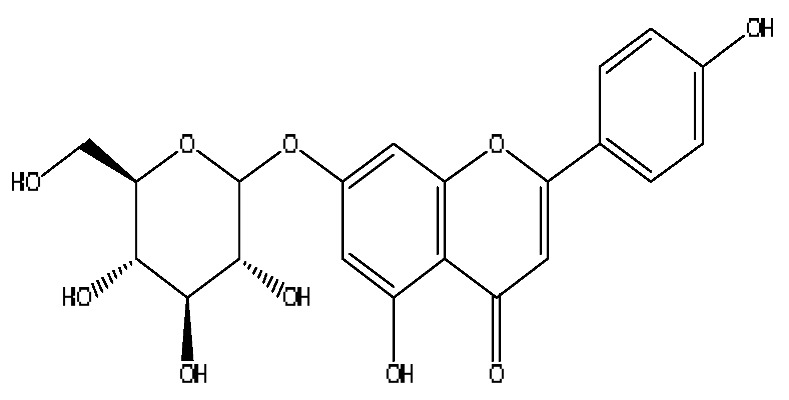
28	Taurohyocholic acid	10.22	514.28	C_26_H_45_NO_7_S	514.28[M−H]^−^,498.29[M−H−O]^−^, 464.30[M−H−CH_6_O_2_]^−^, 304.92[M−H−C_7_H_16_NO_4_S]^−^	*Sus scrofadomestica* Brisson, Bovis calculus Artifactus	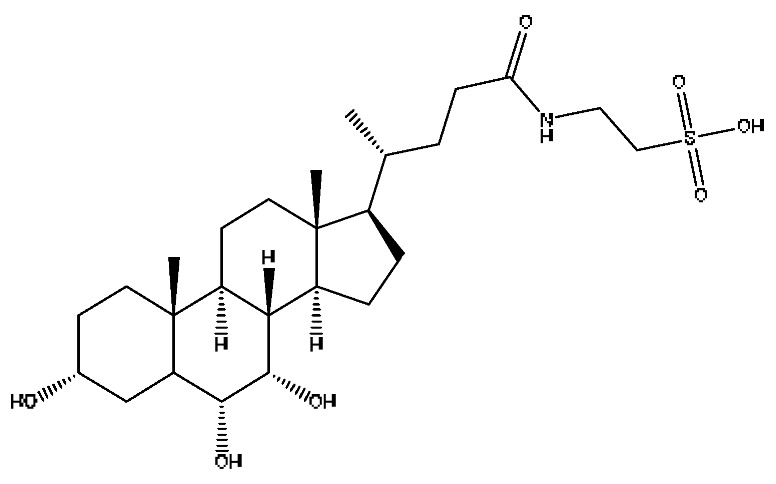
29	Taurocholic acid ^#^	11.46	514.28	C_26_H_45_NO_7_S	514.28[M−H]^−^,462.29[M−H−H_4_O_3_]^−^, 369.23[M−H−C_2_H_9_O_5_S]^−^	Bovis calculus Artifactus	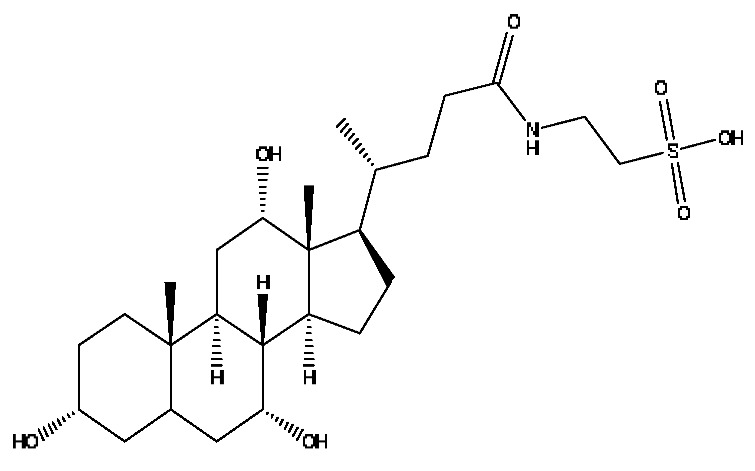
30	Glycohyocholic acid	12.39	464.30	C_26_H_43_NO_6_	464.30[M−H]^−^,405.17[M−H−C_2_H_3_O_2_]^−^, 369.2292[M−H−C_2_H_7_O_4_]^−^	*Sus scrofadomestica* Brisson	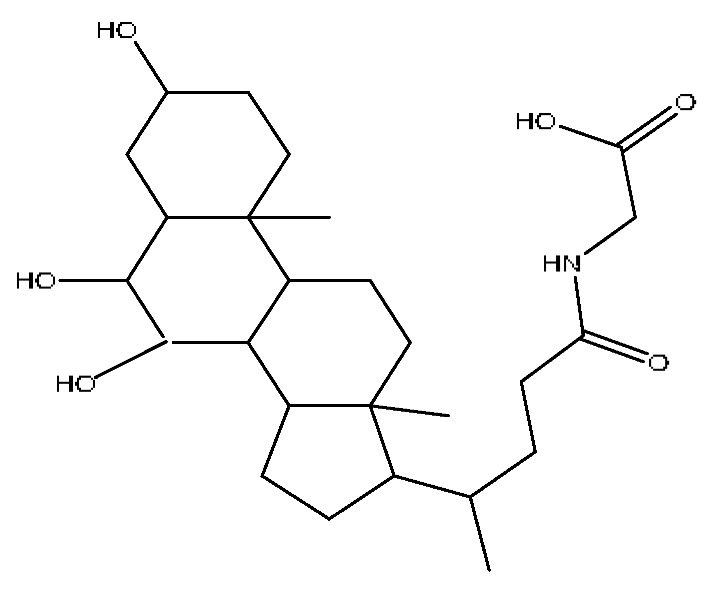
31	Ginsenoside Rb_1_ *^,#^	12.69	1109.61	C_54_H_92_O_23_	1109.61[M+H]^+^,874.44[M+H−C_10_H_19_O_6_]^+^, 786.62[M+H−C_12_H_19_O_10_]^+^	*Panax notoginseng* (Burk.) F.H.Chen	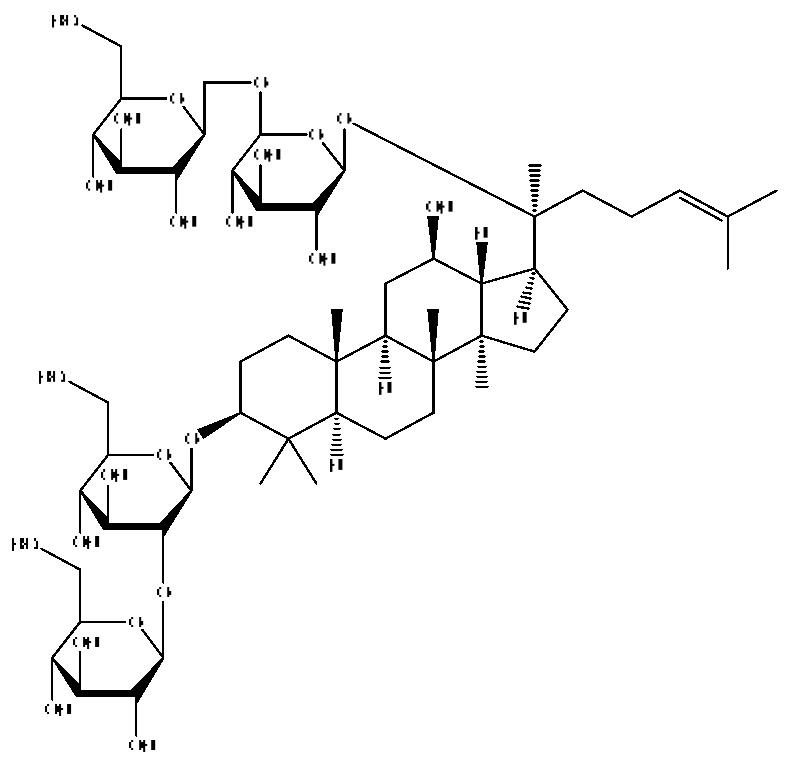
32	Notoginsenoside Fa *^,#^	12.97	1239.55	C_59_H_100_O_27_	1239.54[M−H]^−^,1163.57[M−H−C_3_H_8_O_2_]^−^, 219.84[M−H−C_51_H_88_O_20_]^−^	*Panax notoginseng* (Burk.) F.H.Chen	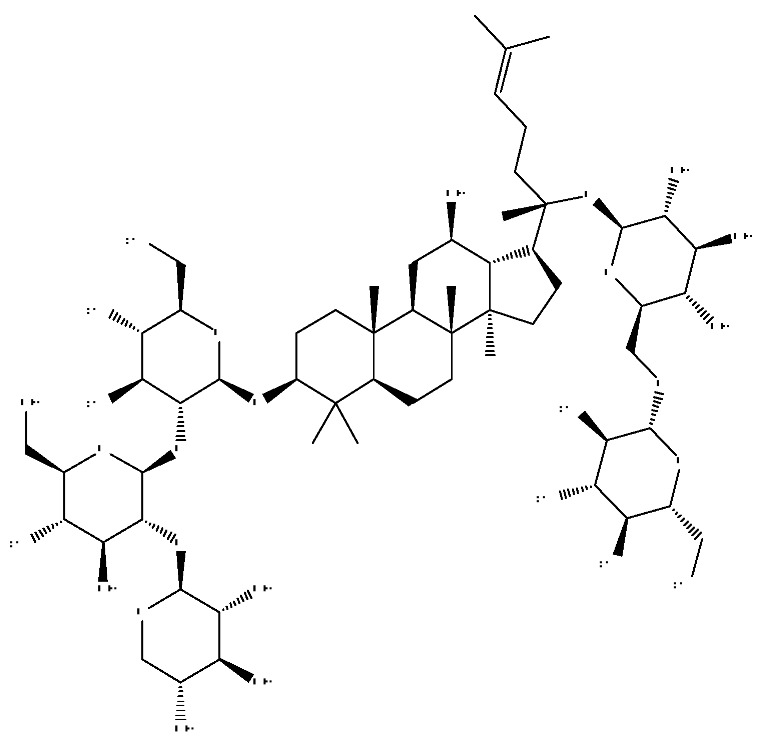
33	Glycocholic acid *	13.54	466.32	C_26_H_43_NO_6_	466.32[M+H]^+^,448.31[M+H−H_2_O]^+^, 430.30[M+H−H_4_O_2_]^+^,412.29[M+H−H_6_O_3_]^+^	*Sus scrofadomestica* Brisson, Bovis calculus Artifactus	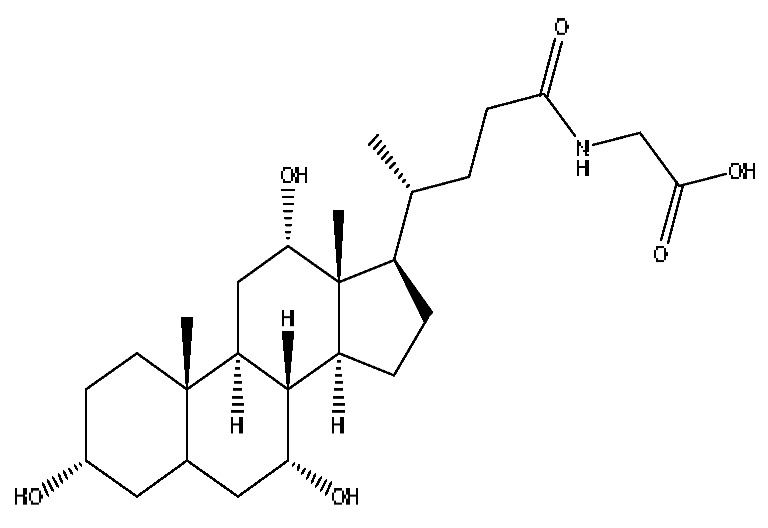
34	Beta-Ionone^#^	13.85	237.15	C_13_H_20_O	237.15[M+FA−H]^−^,221.84[M+FA−H−O]^−^, 195.81[M+FA−H−C_2_H_2_O]^−^	*Lycium barbarum* L.	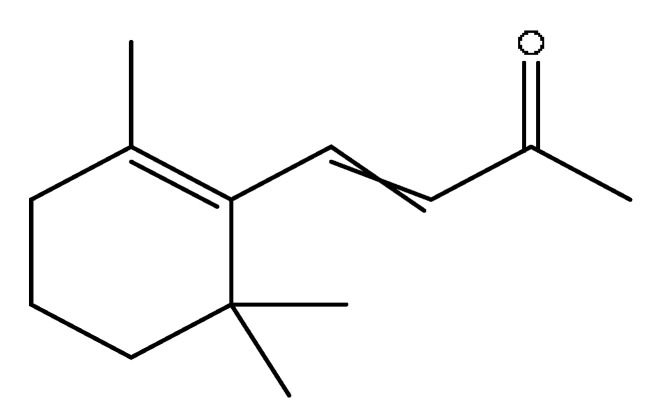
35	Taurohyodeoxycholic acid sodium salt	13.94	522.29	C_26_H_44_NNaO_6_S	522.29[M+H]^+^,343.30[M+H−C_3_H_10_O_4_NNaS]^+^	*Sus scrofadomestica* Brisson, Bovis calculus Artifactus	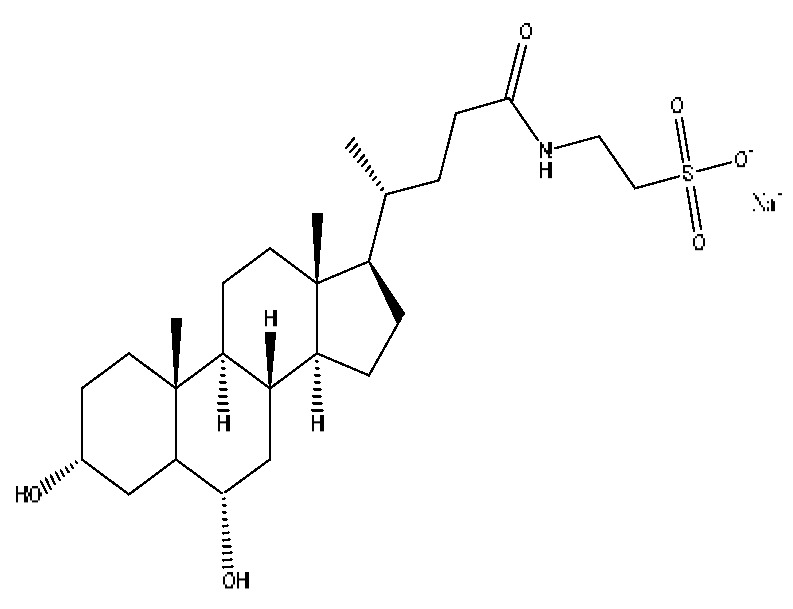
36	Taurohyodeoxycholic acid *^,#^	13.97	498.29	C_26_H_45_NO_6_S	498.29[M−H]^−^,400.23[M−H−H_2_SO_4_]^−^, 329.23[M−H−C_5_H_13_O_4_S]^−^	Bovis calculus Artifactus	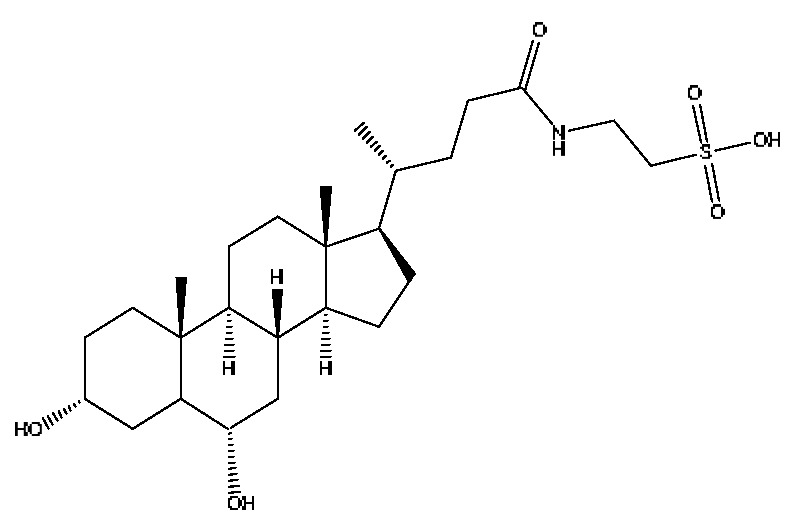
37	Taurochenodeoxycholic acid ^#^	14.59	498.29	C_26_H_45_NO_6_S	498.29[M−H]^−^,465.33[M−H−CH_5_O]^−^, 448.31[M−H−CH_6_O_2_]^−^, 255.82[M−H−C_7_H_17_NO_6_S]^−^	*Sus scrofadomestica* Brisson	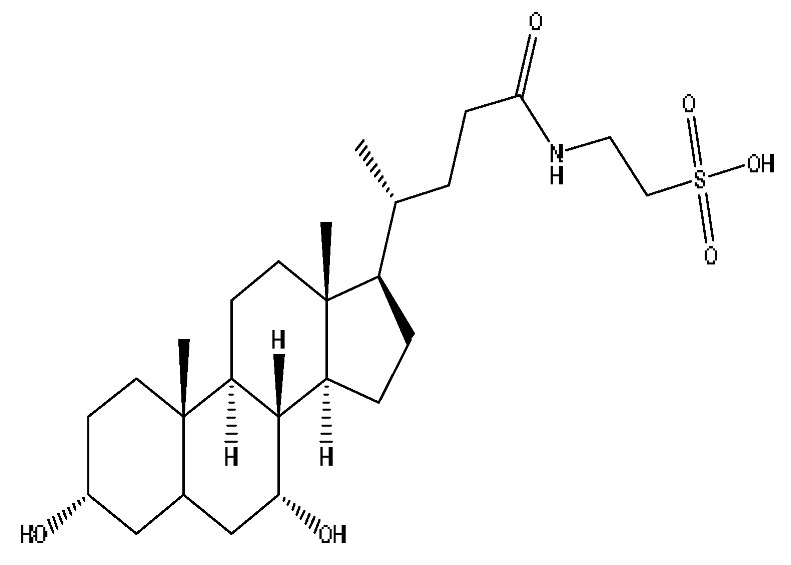
38	Hyocholic acid	15.35	453.29	C_24_H_40_O_5_	453.29[M+FA−H]^−^,407.28[M−H]^−^, 359.19[M−H−H_6_O_3_]^−^,311.22[M−H−C_2_H_8_O_4_]^−^	*Sus scrofadomestica* Brisson, Bovis calculus Artifactus	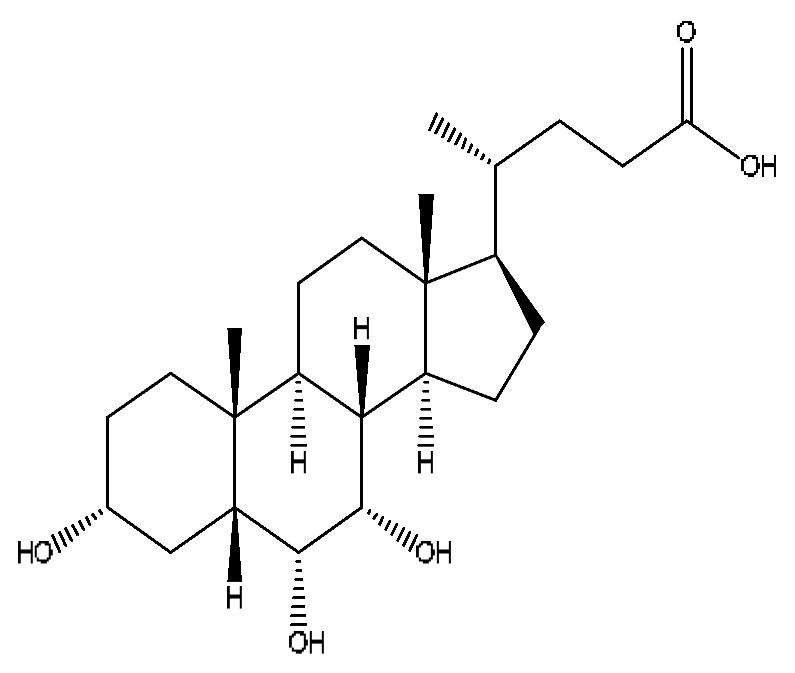
39	Soyasaponin I ^#^	15.75	987.52	C_48_H_78_O_18_	987.52[M+FA−H]^−^,473.32[M+FA−H−C_20_H_34_O_15_]^−^, 437.29[M+FA−H−C_20_H_38_O_17_]^−^	*Abrus cantoniensis* Hance	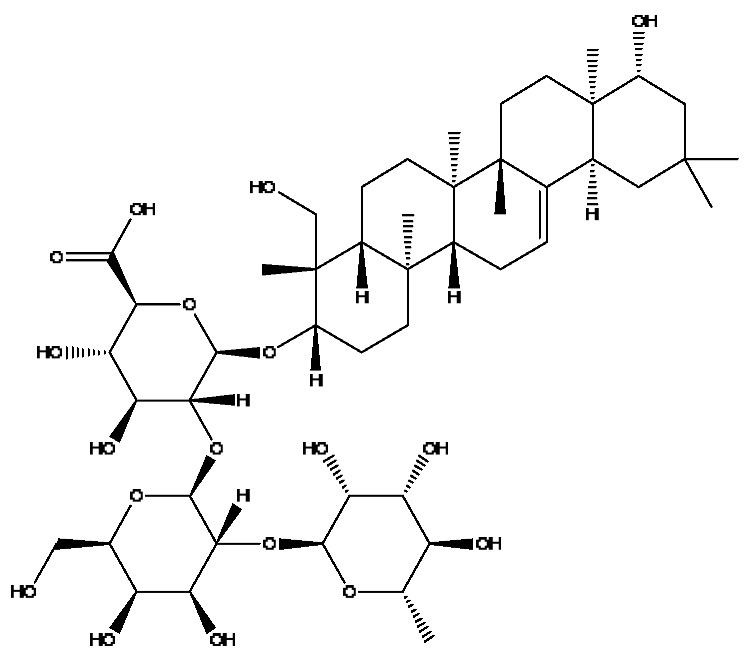
40	Ginsenoside Rh_4_ *^,#^	18.14	665.43	C_36_H_60_O_8_	665.43[M+FA−H]^−^,489.34[M+FA−H−C_7_H_12_O_5_]^−^	*Panax notoginseng* (Burk.) F.H.Chen	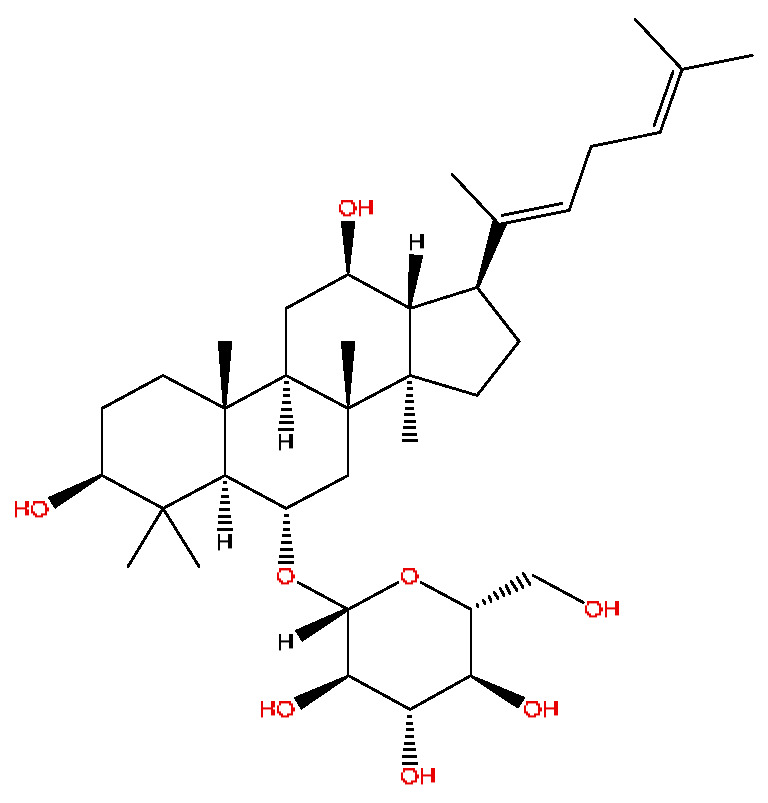
41	Chenodeoxycholic acid *^,#^	20.51	391.29	C_24_H_40_O_4_	391.29[M−H]^−^,297.15[M−H−C_2_H_6_O_4_]^−^, 279.20[M−H−C_6_H_8_O_2_]^−^,261.18[M−H−C_6_H_10_O_3_]^−^	*Sus scrofadomestica* Brisson	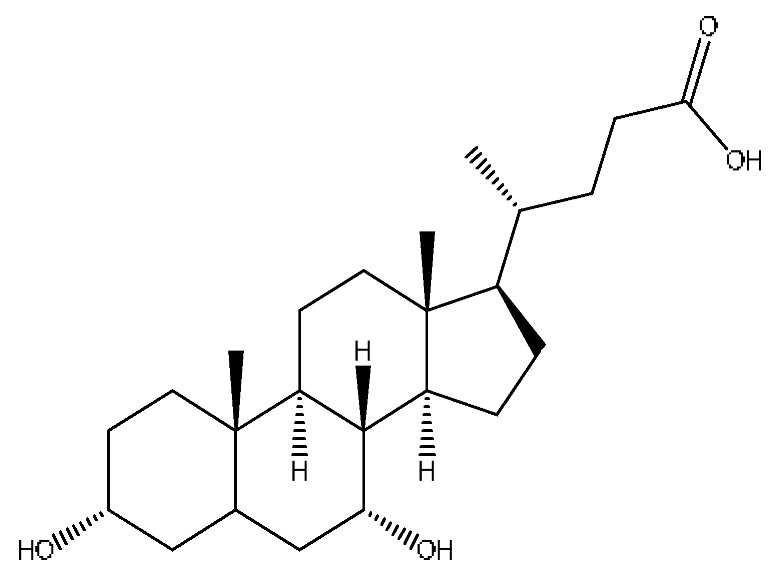
42	Deoxycholic acid *	21.04	391.29	C_24_H_40_O_4_	391.29[M−H]^−^,337.24[M−H−H_6_O_3_]^−^, 297.15[M−H−C_2_H_6_O_4_]^−^,279.20[M−H−C_6_H_8_O_2_]^−^	*Sus scrofadomestica* Brisson	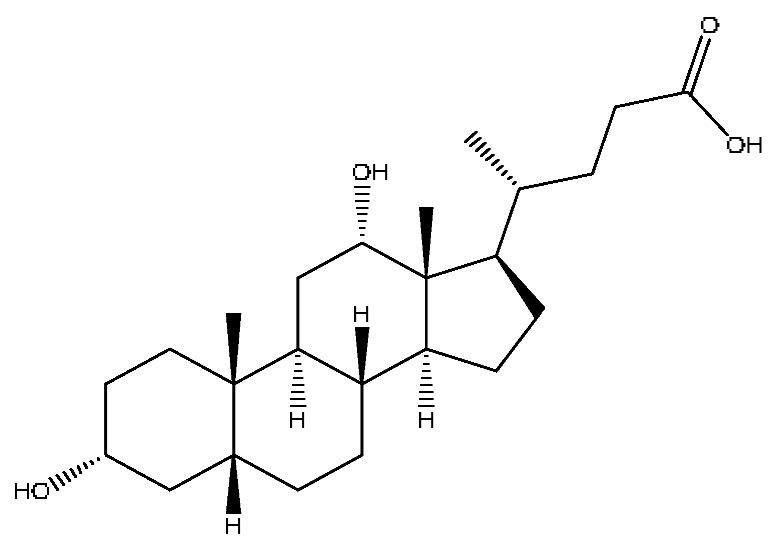
43	Betulonic acid ^#^	24.41	477.34	C_30_H_46_O_3_	477.34[M+Na]^+^,459.25[M+Na−H_2_O]^+^, 441.32[M+Na−H_4_O_2_]^+^	*Ziziphus jujuba* Mill	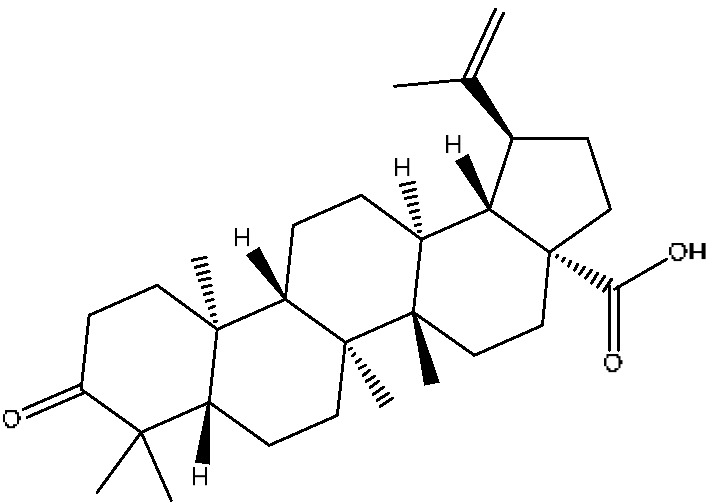

* Compared with reference substance; ^#^ after consulting the literature, it has an obvious liver protection effect.

**Table 2 molecules-28-02494-t002:** The metabolites of JGCC found in the blood.

NO	Metabolite Name	Metabolic Way	Rt	Observed *m*/*z*	Molecular Formula	Ion Form	MS/MS
M_1_	Abrine deoxidized and hydrogenated metabolites	Abrine−O+H_2_	3.44	205.13	C_12_H_16_N_2_O	[M+H]^+^	188.11[M−O]^+^, 146.11[[M−C_2_H_4_NO]^+^
M_2_	Abrine hydroglucuronic acid conjugate	Abrine+H_2_+H_2_O+C_6_H_8_O_6_	3.91	437.15	C_18_H_26_N_2_O_9_	[M+Na]^+^	417.14[M+Na−H_4_O]+, 262.15[M−C8H10NO6]^+^
M_3_	Geniposide oxidized metabolite	Geniposide+2x(+O)	2.00	443.12	C_17_H_24_O_12_	[M+Na]^+^	401.48[M−H_3_O]^+^, 340.14[M−C_2_H_8_O_3_]^+^
M_4_	Geniposide deglycosylated and desaturated metabolite	Geniposide−C_6_H_10_O_5_−H_2_	9.83	225.07	C_11_H_12_O_5_	[M+H]^+^	179.08[M−CHO_2_]^+^
M_5_	Geniposide sulfated metabolite	Geniposide+SO_3_	5.40	513.09	C_17_H_24_O_13_S	[M+FA−H]^−^	245.05[M−C_7_H_11_O_8_]^−^, 165.09[M−C_7_H_11_O_11_S]^−^
M_6_	Geniposide desaturated glucuronic acid conjugate	Geniposide−CH_2_O+2x(−H_2_) +C_6_H_8_O_6_	2.66	529.12	C_22_H_26_O_15_	[M−H]^−^	233.04[M−C_10_H_16_O_10_]^−^
M_7_	Geniposide oxidative hydrogenated glycosylation metabolite	Geniposide+O+H_2_+C_6_H_10_O_5_	3.91	567.20	C_23_H_36_O_16_	[M−H]^−^	241.12[M−C_13_H_27_O_9_]^−^
M_8_	Afrormosin oxidized and desaturated metabolites	Afrormosin+O−H_2_	14.07	357.06	C_17_H_12_O_6_	[M+FA−H]^−^	283.17[M−C_2_H_5_]^−^
M_9_	Afrormosin desaturated metabolite	Afrormosin+2x(−H_2_)	9.44	339.05	C_17_H_10_O_5_	[M+FA−H]^−^	257.82[M−H_5_O_2_]^−^, 146.96[M−C_10_H_12_O]^−^
M_10_	Afrormosin acetylated metabolite	Afrormosin−CH_2_O+2x(+H_2_) +C_2_H_2_O	17.92	337.10	C_18_H_18_O_5_	[M+Na]^+^	253.18[M−C_2_H_5_O_2_]^+^, 191.22[M−C_7_H_7_O_2_]^+^
M_11_	Ginsenoside Rg_3_	Ginsenoside Rb_1_−C_12_H_20_O_10_	25.65	807.49	C_42_H_72_O_13_	[M+Na]^+^	572.38[M−C_8_H_20_O_6_]^+^, 510.37[M−C_13_H_22_O_6_]^+^
M_12_	Ginsenoside Rd deoxymetabolite	Ginsenoside Rb_1_−C_6_H_10_O_6_	21.91	975.55	C_48_H_82_O_17_	[M+FA−H]^−^	476.28[M−C_21_H_42_O_10_]^−^, 279.23[M−C_30_H_51_O_15_]^−^
M_13_	Hyodeoxycholic acid deoxysulfate metabolite	Hyodeoxycholic acid−O+SO_3_	12.81	479.24	C_24_H_40_O_6_S	[M+Na]^+^	409.22[M−C_2_H_7_O]^+^, 393.24[M−CH_3_O_3_]^+^
M_14_	Hyodeoxycholic acid oxidized metabolites	Hyodeoxycholic acid+2x(+O)	15.63	423.28	C_24_H_40_O_6_	[M−H]^−^	405.27[M−H_3_O]^−^, 335.23[M−C_4_H_9_O_2_]^−^
M_15_	Hyodeoxycholic acid hydroglucuronized conjugate	Hyodeoxycholic acid−O+H_2_O+C_6_H_8_O_6_	5.08	615.34	C_30_H_50_O_10_	[M+FA−H]^−^	348.19[M−C_10_H_22_O_5_]^−^, 200.13[M−C_19_H_30_O_7_]^−^
M_16_	Cholic acid desaturated oxidation metabolite	Cholic acid−H_2_+O	26.82	423.27	C_24_H_38_O_6_	[M+H]^+^	323.26[M−C_4_H_3_O_3_]^+^, 240.14[M−C_11_H_18_O_2_]^+^, 184.12[M−C_14_H_22_O_3_]^+^
M_17_	Cholic acid dehydrated glucuronic acid conjugate	Cholic acid−H_2_O+C_6_H_8_O_6_	10.95	589.30	C_30_H_46_O_10_	[M+Na]^+^	504.27[M−C_2_H_6_O_2_]^+^, 488.31[M−CH_2_O_4_]^+^
M_18_	Cholic acid desaturated glucuronic acid conjugate	Cholic acid−H_2_+C_6_H_8_O_6_	21.61	627.31	C_30_H_46_O_11_	[M+FA−H]^−^	526.31[M−C_2_O_2_]^−^, 466.30[M−C_4_H_4_O_4_]^−^
M_19_	Chenodeoxycholic acid dehydrated and sulfated metabolite	Chenodeoxycholic acid−H_2_O+SO_3_	3.94	455.25	C_24_H_38_O_6_S	[M+H]^+^	281.13[M−C_9_H_17_O_3_]^+^, 262.15[M−C_9_H_20_O_4_]^+^, 195.05[M−C_14_H_27_O_4_]^+^
M_20_	Ginsenoside Rg_1_ desaturated metabolites	Ginsenoside Rg_1_+2x(−H_2_)	24.93	797.46	C_42_H_68_O_14_	[M+H]^+^	522.37[M−C_12_H_18_O_7_]^+^, 504.36[M−C_12_H_20_O_8_]^+^, 184.12[M−C_32_H_52_O_11_]^+^
M_21_	Ginsenoside Rg_1_ oxidized metabolite	Ginsenoside Rg_1_+2x(+O)	27.51	855.48	C_42_H_72_O_16_	[M+Na]^+^	546.36[M−C_10_H_22_O_9_]^+^, 487.29[M−C_17_H_29_O_7_]^+^, 323.26[M−C_22_H_37_O_13_]^+^
M_22_	Ginsenoside Rg_1_ oxidative sulfated metabolite	Ginsenoside Rg_1_+O+SO_3_	4.87	919.42	C_42_H_72_O_18_S	[M+Na]^+^	728.35[M−C_6_H_16_O_5_]^+^, 547.30[M−C_15_H_25_O_9_]^+^, 327.15[M−C_21_H_45_O_15_S]^+^
M_23_	Ginsenoside Rg_1_ oxidized glucuronic acid conjugate	Ginsenoside Rg_1_+O+C_6_H_8_O_6_	22.55	1037.52	C_48_H_80_O_21_	[M+FA−H]^−^	476.28[M−C_21_H_38_O_14_]^−^, 396.09[M−C_34_H_60_O_8_]^−^, 279.23[M−C_31_H_49_O_19_]^−^
M_24_	Ginsenoside Rd oxidized metabolite	Ginsenoside Rd+O_2_	22.16	995.54	C_48_H_82_O_21_	[M+H]^+^	522.29[M−C_24_H_40_O_9_]^+^, 494.34[M−C_20_H_36_O_14_]^+^
M_25_	Ginsenoside Rd glucuronic acid conjugate	Ginsenoside Rd+C_6_H_8_O_6_	23.25	1139.58	C_54_H_90_O_25_	[M+H]^+^	570.37[M−C_20_H_40_O_18_]+, 544.34[M−C_22_H_42_O_18_]^+^, 481.32[M−C_27_H_45_O_18_]^+^
M_26_	Ginsenoside Rd deglycosylated oxidation metabolite	Ginsenoside Rd−C_12_H_20_O_10_+2x(+O)	13.62	693.42	C_32_H_62_O_11_	[M+Na]^+^	472.32[M−C_7_H_18_O_6_]^+^, 432.33[M−C_9_H_18_O_7_]^+^, 414.33[M−C_9_H_20_O_8_]^+^, 339.30[M−C_15_H_23_O_8_]^+^
M_27_	Ginsenoside Rb_1_ dehydrated metabolite	Ginsenoside Rd−H_2_O+C_6_H_8_O_6_	22.27	1165.57	C_54_H_88_O_24_	[M+FA−H]^−^	588.33[M−C_27_H_48_O_10_]^−^, 544.27[M−C_28_H_48_O_12_]^−^, 524.28[M−C_30_H_44_O_12_]^−^
M_28_	Notoginsenoside T_5_ desaturated metabolite	Notoginsenoside T_5_−H_2_	27.41	751.46	C_41_H_66_O_12_	[M+H]^+^	482.35[M−C_14_H_20_O_5_]^+^, 464.34[M−C_10_H_22_O_9_]^+^, 381.33[M−C_14_H_25_O_11_]^+^
M_29_	Notoginsenoside T_5_ oxidized glucuronic acid conjugate	Notoginsenoside T_5_−C_5_H_8_O_5_+2x(+O)+C_6_H_8_O_6_	27.19	813.47	C_42_H_68_O_15_	[M+H]^+^	546.36[M−C_10_H_18_O_8_]^+^, 524.39[M−C_12_H_16_O_8_]^+^, 481.33[M−C_15_H_23_O_8_]^+^, 381.33[M−C_15_H_27_O_14_]^+^
M_30_	Notoginsenoside T_5_ dehydrated glucuronic acid conjugate	Notoginsenoside T_5_−C_5_H_8_O_5_+2x(−H_2_O)+C_6_H_8_O_6_	4.44	743.43	C_42_H_64_O_11_	[M−H]^−^	245.05[M−C_29_H_55_O_6_]^−^, 165.09[M−C_32_H_64_O_12_]^−^
M_31_	Scoparone hydrosulfated metabolite	Scoparone−CH_2_+H_2_+SO_3_	5.56	273.01	C_10_H_10_O_7_S	[M−H]^−^	257.82[M−OH]^−^, 193.03[M−HSO_3_]^−^
M_32_	Scoparone hydrogenated hydroxylation metabolite	Scoparone−CH_2_O+2x(+H_2_O)	2.13	211.06	C_10_H_12_O_5_	[M−H]^−^	197.81[M−CH_3_]^−^, 123.04[M−C_3_H_5_O_3_]^−^
M_33_	Capillarisin hydrogenated metabolite	Capillarisin+H_2_	2.94	317.07	C_16_H_14_O_7_	[M−H]^−^	203.08[M−C_6_H_11_O_2_]^−^, 172.99[M−C_7_H_14_O_3_]^−^

**Table 3 molecules-28-02494-t003:** Regression equation, linear range, LOD and LOQ of the 16 active ingredients in JGCC.

Number	Compounds	Regression Equation	R^2^	Linear Range (ng/mL)	LOD (ng/mL)	LOQ (ng/mL)
1	Trigonelline	Y = 2227X + 781,700	0.9993	12.18~24,360	0.12	0.61
2	Abrine	Y = 132X − 4611	1.0000	9.98~99,800	0.50	2.50
3	Hypaphorine	Y = 8061X + 2,141,000	0.9997	15.28~30,560	0.15	0.76
4	Genipin-1-gentiobioside	Y = 587.9X + 40,930	0.9999	10.22~20,440	0.10	0.51
5	Geniposide	Y = 25.5X + 2832	0.9999	13.89~138,900	0.14	0.69
6	Vicenin-2	Y = 399.6X + 10,760	1.0000	7.21~72,100	0.07	0.36
7	Albiforin	Y = 66.1X + 799.2	1.0000	10.31~103,100	0.10	0.52
8	Paeoniflorin	Y = 2.5X + 178.7	0.9999	115.1~115,100	2.88	115.10
9	Isoschaftoside	Y = 384.1X + 59,040	0.9997	10.93~109,300	0.11	0.55
10	Isovitexin	Y = 1147X − 1774	0.9999	10.76~2690	0.11	0.54
11	Ginsenoside Rg_1_	Y = 254.5X + 19,910	0.9999	10.93~21,860	0.11	0.55
12	Luteolin	Y = 1607X + 10,180	0.9998	11.16~2790	0.11	0.56
13	Taurohyodeoxycholic acid	Y = 614.9X − 635	1.0000	10.98~21,960	0.11	0.55
14	Notoginsenoside Fa	Y = 28.1X + 491	0.9999	105.1~21,020	10.51	105.10
15	Ginsenoside Rb_1_	Y = 7.3X + 1187	0.9998	11.02~22,040	2.76	11.02
16	Chenodeoxycholic acid	Y = 9.4X − 916.9	0.9999	10.50~21,000	0.53	2.63

In the regression equation Y = aX + b, X is the concentration, Y is the peak area, R is the correlation coefficient of the equation.

**Table 4 molecules-28-02494-t004:** Determination results of JGCC in 14 different batches.

Compounds	2111095	2105033	2111092	2111093	2111094	2102006	2102007	2102008	2101005	2101003	2111091	2111099	2108070	2108069	Average (mg/g)	SD (mg/g)	RSD (%)
Trigonelline (**1**)	2.04	1.39	1.67	1.72	2.10	1.85	1.80	1.81	1.51	1.80	2.31	2.08	1.68	1.54	1.81	0.25	14.09
Abrine (**2**)	2.72	2.64	5.24	4.07	3.87	3.88	1.17	5.09	3.72	5.87	4.39	3.48	6.48	6.42	4.22	1.51	35.78
Hypaphorine (**3**)	0.31	0.30	0.26	0.26	0.36	0.27	0.25	0.30	0.26	0.29	0.37	0.36	0.35	0.25	0.30	0.04	14.57
Genipin-1-gentiobioside (**4**)	2.58	2.79	3.01	2.79	2.66	2.70	2.41	2.69	2.47	2.40	2.72	2.66	2.79	2.91	2.68	0.18	6.60
Geniposide (**5**)	5.17	5.18	5.91	5.88	5.31	5.40	4.99	5.28	5.08	5.32	5.34	5.63	5.41	5.64	5.40	0.28	5.15
Vicenin-2 (**6**)	1.70	1.45	1.55	1.62	1.23	1.64	1.44	1.36	1.77	1.51	1.28	1.42	1.35	2.01	1.52	0.21	13.89
Albiforin (**7**)	1.34	1.70	1.67	1.60	1.76	1.74	1.70	1.88	1.72	1.70	1.79	1.72	1.69	1.70	1.69	0.12	7.09
Paeoniflorin (**8**)	2.62	2.59	2.95	2.92	2.59	3.09	2.71	3.34	2.82	2.77	2.89	2.74	3.43	3.49	2.93	0.31	10.44
Isoschaftoside (**9**)	4.31	3.71	4.11	3.73	3.60	4.35	4.20	3.76	4.80	4.16	3.65	4.17	3.06	4.84	4.03	0.48	11.98
Isovitexi (**10**)	0.07	0.06	0.07	0.06	0.05	0.07	0.07	0.06	0.09	0.07	0.05	0.06	0.05	0.08	0.07	0.01	17.18
Ginsenoside Rg_1_ (**11**)	1.51	2.10	1.90	1.60	1.89	2.03	1.54	1.92	1.69	1.80	1.76	1.77	2.10	2.10	1.84	0.20	11.14
Luteolin (**12**)	0.09	0.11	0.08	0.07	0.07	0.10	0.11	0.10	0.09	0.08	0.07	0.07	0.06	0.10	0.08	0.02	18.17
Taurohyodeoxycholic acid (**13**)	0.42	0.52	0.46	0.44	0.48	0.55	0.50	0.58	0.53	0.56	0.46	0.47	0.43	0.47	0.49	0.05	10.33
Notoginsenoside Fa (**14**)	0.06	0.09	0.06	0.04	0.04	0.06	0.04	0.05	0.06	0.06	0.07	0.06	0.06	0.08	0.06	0.01	24.52
Ginsenoside Rb_1_ (**15**)	1.24	1.95	1.45	1.34	1.32	1.60	1.12	1.36	1.31	1.21	1.14	1.23	1.45	1.37	1.36	0.21	15.57
Chenodeoxycholic acid (**16**)	0.84	1.74	0.90	0.80	0.85	1.45	1.32	0.66	1.31	1.27	0.90	0.78	2.15	1.88	1.20	0.46	38.56

## Data Availability

All data are available in the Appendix A and this paper.

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
