# Peer review of "Quality Markers’ Discovery and Quality Evaluation of Jigucao Capsule Using UPLC-MS/MS Method"

_molecules, 2023, doi:10.3390/molecules28062494_

Round 1

Reviewer 1 Report (Previous Reviewer 1)

Overall, this manuscript still has numerous grammatical errors throughout. I have not made reference to them all as there are far too many. Hence, I would advise the authors to avail themselves to a professional editing service to improve the manuscript. With regards to the methods and description of the animal groups (section 2.6), this is a tad confusing and hence, needs to be written more concisely with the animal groups definitively defined. Section 3.4 does not read like a results section. In fact, it reads as it was written for the discussion. Also, the authors have still not provided an explanation/overview of the results in figures 2-4.

My general comments are as follows:

Abstract:

- line 27: please delete the 'comma' after the word 'stage'.

-line 29: no need to capitalise 'Ultra', please amend to lower case.

-line 32: please define 'RSD'.

Introduction:

-line 39: 'Hance' should not be italicised. Please amend entire manuscript accordingly.

-lines 40 and 41: please insert space before references '[1], [2,3].

-lines 43 and 44: 'Ellis, Pall, Mill and Busson' should not be italicised. Please amend entire manuscript accordingly.

-line 76: please replace 'a' with the word 'an'.

-line 77 sentence beginning with (SBW): 'The pharmacodynamic' is grammatically incorrect and the sentence is far too long. Please amend.

-line 78: please delete 's' from the word 'terms'.

-line 83: there is a word missing between 'as' and 'was. Please amend.

Materials and Methods:

-line 144: the word 'via' is Latin and hence by convention, should be italicised. Please amend.

-line 117: no need to italicise 'Rosc.' please amend.

-line 152, SBW: 130 mg, is grammatically incorrect. Please amend.

-line 155: what temperature did you boil the aqueous extract? please include details. Also, what temperature did you reduce at for the same solution? Please clarify and include details. Further, what was the mixture filtered through? Please clarify.

-line 157: how were the filtrates concentrated? Please clarify.

-line 175: Please define 'HE.'

-lines 183 and 184: What exactly was washed? The column? Please clarify.

-line 239: yes, but how much exactly was added? Please be specific, not just state, low, medium and high. 

Results:

-please amend figures 2-4 inclusive as indicated in comments above.

-please amend section 3.4 as noted above.

-line 372: 'and it was added natriumion' does not make sense. Please amend.

Discussion:

-line 488: never use the term 'etc' in a scientific manuscript. Please delete.

-line 489, SBW: 'Experimental studies', requires referencing, please amend.

-line 490 SBW: 'It also reduces', requires referencing, please amend.

-line 515, SBW: 'A large number of studies', yet you reference only two. Two does not equate to a large number. Please amend.

-line 521, SBW: 'Although this study', which study are you referring to? Please clarify.

Author Response

Dear reviewer:

Thank you very much for your attention and timely response to our article. For the questions raised by reviewers, we make the following replies.

Abstract:

  1. line 27: The comma after "stage" has been deleted, see line 27.
  2. line 29: "Ultra" has been changed to lower case, see line 29.
  3. line 32: "RSD" has been defined, see line 32.

Introduction:

  1. line 39: "Hance" has been changed to non-italic, as shown in the entire manuscript.
  2. line 40 and 41: A space has been inserted before references [1], [2,3].
  3. line 43 and 44: "Ellis, Pall, Mill and Busson" has been changed to non-italicized, as shown in the entire manuscript.
  4. line 76: The whole sentence has been amended, see line 80-86.
  5. line 77: Sentence beginning with (SBW):"The pharmacodynamic" has been amended, see line80-86。
  6. line 78: The whole sentence has been amended, see line80-86。
  7. line 83: The word "it" has been added between "as" and "was", see line 87.

Materials and Methods:

  1. line 114: "via" has been italicized, see line 430.
  2. line 117: "Rosc" has been changed to non-italic, see line432。
  3. line 152: The sentence has been amended, see line 468.
  4. line 155: An induction cooker is used for decocting. After boiling, the cooker is changed to a soft fire and keep it slightly boiling for 1h. The filtrate is filtered with 4 layers of gauze, see line 471-474.
  5. line 157: The filtrates concentratedthrough decocting, see line
  6. line 175: It has beendefined "HE", see line 
  7. lines 183 and 184: The column was washed, see line501.
  8. line 239: See the column "Added amount" in Table S1for the specific added amount.
  9. Section 2.6 changes to section 4.6, animals divided into three groups, named control group, model group and JGCC grouprespectively, the drugs given by each group were shown in the manuscript of line 482-488.

Results:

  1. The results shown in Figure 2-4 have been described, see line116-130, line 131-136 and line 154-156 respectively.
  2. Section 3.4changes to section 2.4, section 2.4 is a result based on section 4.7 of "Materials and Methods" and should be placed in the "Results" 
  3. line 372: The sentence has been amended, see line218-220.

Discussion:

  1. line 488: The sentence has been amended, see line333-335.
  2. line 489: SBW: "Experimental studies", referencinghas been added, see line
  3. line 490: SBW: "It also reduces", referencinghas been added, see line
  4. line 515: SBW: "A large number of studies"has been amended, see line 
  5. line 521: SBW:"Although this study" has been amended, see line 370-371.

Reviewer 2 Report (Previous Reviewer 2)

The current version of the manuscript has been greatly improved both in terms of writing and content. The experiments are well designed, the argumentation process adequate, the results reliable and the conclusions convincing. There are also no obvious errors in the details. 

Author Response

Dear reviewer:

      Thanks to your a lot of valuable time reading our article and providing many valuable opinions on the revision of our manuscript.

This manuscript is a resubmission of an earlier submission. The following is a list of the peer review reports and author responses from that submission.

Round 1

Reviewer 1 Report

Overall, I found this manuscript to be grossly under referenced (specific instances are provided below). Grammatical errors were made throughout, however, section 3.3 was very poorly written. With regards to the tables and figures, the presented data was not adequately described. All tables and figures should be replete with a proper overview of the results as it is not up to the reader to interpret the data. My other concerns regarding this manuscript was discussing data in the results section rather than placing it in the discussion where it can be fully discussed and contextualised (specific instances are below). In addition, I feel that the discussion itself is lacking depth and requires greater contextualisation of the results generated in this study compared with the current literature.

General comments:

Abstract:

-line 23 sentence beginning with (SEW): 'To analyse the active' does not make sense. Please amend.

-line 24 SBW: 'On the basis' does not make sense. Please amend.

-line 28: please italicise the words 'in vitro'

Introduction:

-line 41 SBW: 'The monarch', requires referencing. Please amend.

-line 43 SBW: 'This was recorded as' does not belong here as I do not believe it is appropriate for this journal. Please delete.

-line 56: please insert a full stop after the word' jaundice' and delete the rest of the sentence. Additionally, the sentence requires referencing. Please amend.

-lines 47-48: 'It has a long history', requires referencing. Please amend.

-lines 50-53: please italicise all plant names.

-line 55: please insert a comma after the word 'hepatitis B'

-line 57: please insert a space between the full stop and 'JGCC'

-line 57 SBW: 'JGCC can also assist', requires referencing. Please amend.

-line 61 SBW: 'It can also reduce', requires referencing. Please amend.

-line 63 SBW: ''It additionally increases', requires referencing. Please amend.

-line 68: please insert a space between 'biosynthesis' and the bracket.

-line 69 SBW: 'Although JGCC', requires multiple referencing. Please amend.

-line 71 SBW: 'The current standards', requires referencing. Please amend.

-line 72: please italicise plant name.

-line 82: please write TCM in full, then abbreviate.

-line 84: please insert space between 'proposed' and the bracket.

-line 85 SBW: 'After several years', requires referencing. Please amend.

-line 88: please insert space between 'TCM' and the bracket.

-line 89 SBW: 'Effectiveness and safety', is grammatically incorrect. Please amend.

-line 93: please insert space between 'TCM' and the bracket.

-line 98: please insert space between the full stop and the word 'Serum'

-lines 93-97: What exactly are you trying to say in this section? Please be concise and amend accordingly.

-lines 100-109: Same comment as directly above. 

-line 109: please insert space between 'process' and the bracket.

-line 110 SBW: 'DHJS', requires referencing. Please amend..

-line 117: please insert space between 'constituents' and the bracket.

Materials and Methods:

- was a voucher specimen deposited in an appropriate herbarium? Please include voucher number and herbarium details.

-line 160: please define the term SRM and then abbreviate.

-line 168: please add the letter 'd' to the word 'immerse'

-line 175: by convention, all numbers below 10 are written in full, whilst those 10 and above, are written numerically (unless the are the first word in a sentence, then they should be written in full).  Please write '8' in full.

-line 176: please define what an 'SD rat' is.

-line 185: how were the rats sacrificed? Please include the relevant details.

-line 194: please define the term HLB.

-line 197: what temperature was the vacuum concentrator operated at? Please include details.

-line 202: please insert space between 'method' and bracket.

-line 207: what percentage methanol did you use? Please include.

-line 233: why were there three levels? i.e. how much exactly was added? Please include relevant details as to me, this section does not make sense and hence, is not reproducible by an external researcher.

Results:

-line 265: where have you referenced Figure 1 in your text? Please amend.

-line 280: please insert a space between the different types of brackets.

-line 284: please write '9' in full

-line 285: please write '2' in full

-line 286: please write numbers '4' and '5' in full

-line 287: please write numbers '3' and '4' in full

-line 288: please write numbers '4' and '8' in full

-line 289: please write '6' in full

-line 292: please write '4' in full

-line 303 (Table 1): please italicise all plant names

-lines 318-323: this section is not a result. Please move to the discussion.

-line 326: please write +Na in full

-line 329: please replace the word 'is' with 'was'

-line 349 SBW: please delete from 'According to' all the way through to 'Liu', then please capitalise the word 'we'

-line 355: please write '6' in full

-line 355 SBW: 'Among them' does not belong in the results section. Please move to the discussion.

-line 360: whole section starting on this line does not make sense. Please amend.

-line 362: please write '6' in full

-line 371: you should only ever present results and not contextualise what they mean in the results section. Contextualisation should always be reversed for the discussion.

-line 431 Table 4: the way the headings in this table are set out is confusing. Please amend.

Discussion:

-line 450 sentence ending with (SEW): 'swelling pain', requires referencing. Please amend.

-line 451 SEW: 'summer', requires referencing. Please amend.

-line 453: the words 'and so on' are not scientific. Please delete.

-line 454 SBW: 'Trigonelline', requires referencing. Please amend.

-line 459: please define the terms NAFLD and MMP-2/9

-line 462 SEW: 'mice', requires referencing. Please amend.

-line 466: please define terms AST and ALT

-line 467 SEW: 'cholesterol', requires referencing. Please amend.

-line 469 SBW: 'Ginsenoside', requires referencing. Please amend.

-line 470 SBW: 'Ginsenoside', requires referencing. Please amend.

-line 484 SBW: 'Chlorogenic acid', requires referencing. Please amend.

-line 485 SBW: 'A large number', requires multiple referencing. Please amend.

-line 488 SBW: 'Geniposide', requires referencing. Please amend.

-line 489 SBW: 'It can also', requires referencing. Please amend.

-line 491: please insert space between 'mechanism' and bracket

-line 493 SEW: 'effects', requires referencing. Please amend.

-line 497: you state 'their metabolites showed a high response in the serum.' Are you referring to their actual levels in the serum? Please clarify.

-line 500: please define TLR.

-line 502 SEW: 'immunomodulation', requires referencing. Please amend.

-line 502 SBW: 'Studies have shown', studies is plural, and yet you have only cited a single reference. Please amend.

-line 510: please write '2' in full

-line 513 SEW: 'receptor' requires referencing. Please amend.

-lines/sentences 513-515: all require referencing. Please amend.

-line 522 SBW: 'Taurohyodeoxycholic acid', requires referencing. Please amend.

-line 555: please write 33 in full

Reviewer 2 Report

Jigucao capsules (JGCC) may be a herbal preparation with a specific function, and the authors believe that the current drug has a quality deficiency on the grounds that no content determination standards are given. Therefore, the authors developed what the authors believe to be a solution with the help of mass spectrometry based on metabolite analysis. The entire manuscript describes the authors' own story, which does not inspire others to read or think about it, and the manuscript is mediocre. I also find that such a manuscript does nothing to promote the academic community.

The attachments are not well produced and lack the most basic information.

Reviewer 3 Report

This manuscript reports the results of the investigation of constituents or metabolites as possible quality markers of the Jigucao capsules. The results were promising and supported by tables and figures. However, there are a few issues that need to be considered and required correction:

First of all, the Abstract should be more concise and scientifically sound. What do means „the existing quality standard for JGCC has defects“? What are „active components“? How did the authors conclude that all constituents/metabolites detected in blood have pharmacological activities? This should be considered throughout the manuscript.  

Page 1; Line 27: please provide the full name for UNIFI. 

Page 1; Line 40: please consider that JGCC is the product and capsules are the pharmaceutical forms. The first sentence should be rephrased.

Page 2: Lines 50-53: Latin plant names should be present in italic

Page 2: Line 60: What is the status of JGCC capsules? Please explain what does ideal drug means.

Page 2: Line 82: please provide the full name for TCM

The goal of the research should be clearly formulated at the end of the Introduction

Page 12: Lines 318-322: please add reference(s)

Page 19: Lines 446-452:  please add reference(s)

The conclusion section should be extended. What are the benefits and significance of this research? How and by whom can the results of the conducted research be used and for what purpose? Only at least the answers to these questions, taken into account in the abstract, introduction, and conclusions could give the article a more complete character.

Some sentences are hard to understand and they distract from the overall quality of the work. The manuscript would benefit from English editing. 

Reviewer 4 Report

Present study evaluated the methanol soluble components of Jigucao capsule and their major metabolites in the rats’ blood. Several improvements are needed to increase the significance of the study.

Line 146: “0.1% formic acid water (A) and 0.1% formic acid acetonitrile” have an ambiguous meaning.

Line 149: with elution rate of 0.4 mL/min.

Line 181: How is ANT olive solution prepared?
Line 182: Is the preparation of DHJS model rats referred to previous studies? Is there any evaluation index to prove the successes of DHJS model establishment?

Line 197: “uL” should be “μL”.

Line 209: there are 16 active ingredients, why only 5 standard curves are drawn?

Line 225: In what light and temperature conditions the samples were exposed?

Table 1: It is suggested to replace the “Origin” with “References”, and replace the “a…j” with the specific references numbers.

Figure 3: The peak of abrine is not clearly shown in figure A, it will be better to point it with arrow. How is the response value in figure B calculated? It is not stated in the manuscript. “control, model” should be “control model”. The parameters for low-energy collision and high-energy collision should be accurately stated.

Table 2: The metabolites in the serum should be chemically identified and the name of the metabolic product should be given.

Line 349~352: The determination principles of Q-markers should be described in more detail in the Method section.

In addition, the results did not show the effectiveness of ingredients, thus, how to insure the accuracy and dependability of the compounds as Q –markers?